# Differential compartmentalization of myeloid cell phenotypes and responses towards the CNS in Alzheimer's disease

Camila Fernández Zapata [1,2,3], Ginevra Giacomello [4], Eike J. Spruth[3,5], Jinte Middeldorp[6,7], Gerardina Gallaccio [1,2,3], Adeline Dehlinger [1,2,3], Claudia Dames [8], Julia K. H. Leman[3,9], Roland E. van Dijk[6], Andreas Meisel [8,10], Stephan Schlickeiser [11], Desiree Kunkel[12], Elly M. Hol [6], Friedemann Paul[1,2,8,10], Maria Kristina Parr [4], Josef Priller [3,5,13,14,15] ✉ & Chotima Böttcher [1,2,3,15] ✉

Myeloid cells are suggested as an important player in Alzheimer´s disease (AD). However, its continuum of phenotypic and functional changes across different body compartments and their use as a biomarker in AD remains elusive. Here, we perform multiple state-of-the-art analyses to phenotypically and metabolically characterize immune cells between peripheral blood ($n = 117$), cerebrospinal fluid (CSF, $n = 117$), choroid plexus (CP, $n = 13$) and brain parenchyma ($n = 13$). We find that CSF cells increase expression of markers involved in inflammation, phagocytosis, and metabolism. Changes in phenotype of myeloid cells from AD patients are more pronounced in CP and brain parenchyma and upon in vitro stimulation, suggesting that AD-myeloid cells are more vulnerable to environmental changes. Our findings underscore the importance of myeloid cells in AD and the detailed characterization across body compartments may serve as a resource for future studies focusing on the assessment of these cells as biomarkers in AD.

Alzheimer´s disease (AD) is the most common neurodegenerative disease that is recognized as one of the top devastating diseases worldwide, due to its high costs caused by patient care and management. The amyloid cascade hypothesis, a widely accepted hypothesis referring to AD pathogenesis, proposes amyloid plaques or the β-amyloid (Aβ)-peptides as the direct cause of progressive neurodegeneration. A cascade initiated by Aβ deposition progressively leads to Tau pathology, synaptic dysfunction, neuronal loss and ultimately dementia[1]. However, collective evidence obtained from patients with familial AD have questioned the linearity of the amyloid cascade in AD

[1]Experimental and Clinical Research Center, a cooperation between the Max Delbrück Center for Molecular Medicine in the Helmholtz Association and Charité Universitätsmedizin Berlin, Berlin 13125, Germany. [2]Max Delbrück Center for Molecular Medicine in the Helmholtz Association (MDC), Berlin 13125, Germany. [3]Department of Neuropsychiatry and Laboratory of Molecular Psychiatry, Charité—Universitätsmedizin Berlin, corporate member of Freie Universität Berlin and Humboldt-Universität zu Berlin, Berlin 10117, Germany. [4]Institute of Pharmacy, Freie Universität Berlin, Berlin 14195, Germany. [5]DZNE, Berlin 10117, Germany. [6]Department of Translational Neuroscience, University Medical Center Utrecht Brain Center, Utrecht University, 3584 CX Utrecht, The Netherlands. [7]Department of Neurobiology and Aging, Biomedical Primate Research Centre, 2288 GJ Rijswijk, The Netherlands. [8]Department of Neurology and Experimental Neurology, Charité-Universitätsmedizin Berlin, Berlin 10117, Germany. [9]Institute of Biology, Humboldt–Universität zu Berlin, Berlin 10115, Germany. [10]NeuroCure Clinical Research Center, Charité-Universitätsmedizin Berlin, Berlin 10117, Germany. [11]Institute of Medical Immunology, BIH Center for Regenerative Therapies (BCRT), Berlin Institute of Health at Charité - Universitätsmedizin Berlin, Berlin 13353, Germany. [12]Flow & Mass Cytometry Core Facility, Berlin Institute of Health at Charité - Universitätsmedizin Berlin, Berlin 13353, Germany. [13]Department of Psychiatry and Psychotherapy, School of Medicine, Technical University Munich, Munich 81675, Germany. [14]UK DRI, University of Edinburgh, Edinburgh EH16 4SB, UK. [15]These authors jointly supervised this work: Josef Priller, Chotima Böttcher. ✉e-mail: josef.priller@charite.de; chotima.boettcher@charite.de

pathology, particularly regarding the gradual evolution of the disease in humans[2–4]. In addition, it is imperative to accommodate the complex compensation mechanisms of multiple cell types in a hypothesis. The empirical observations of (i) microglial activation in the brain[5,6], (ii) increased production of inflammatory mediators in the brain, peripheral blood and cerebrospinal fluid (CSF), which are related to the blood-CSF barrier (BCSFB) disruption[7–9] and (iii) differential myeloid and lymphoid cell responses in the peripheral blood and CSF[10,11] leads to a hotly debated theory of the two poles of AD, (immune-driven) neuroinflammation and neurodegeneration. According to this theory, neurodegeneration could be promoted by microglia/macrophages that respond to an increased inflammatory environment in the pathological brain[12–14], or vice versa neuroinflammation could be initiated by the local neurodegeneration in the brain. Results obtained from studies in mouse models of AD suggest that multiple myeloid cell populations in different body compartments significantly modify the disease outcome via different mechanisms. For example, microglia play a dichotomous role which alternates between protective clearance of β-amyloid and debris, and detrimental neurotoxic effects[14,15]. The CNS-associated macrophages (CAMs) are key immune cells involved in the regulation of cerebral amyloid angiopathy (CAA) and thus modify the disease burden in AD[16–18], whereas the hematogenous myeloid cells[16,19] do not seem to significantly modify the disease outcome in mouse models of AD. However, studies in humans showed that the CD33 AD-risk allele is linked to higher expression of CD33 on monocytes, and an expression quantitative trait locus (eQTL) study in patients with autoimmune or neurodegenerative diseases revealed that the AD susceptibility alleles are significant eQTLs only in monocytes, suggesting an involvement of this cell type in human AD pathology[20,21]. Unlike studies in mouse models, technical limitations of studies using human specimens, especially the difficulties in procuring human specimens from different body compartments, have confined the understanding of the continuum of myeloid cell diversity to functional changes towards the CNS and/or AD pathology. Using mass cytometry, we could demonstrate the phenotypic differences between human immune cells in the CSF and those in the peripheral blood, and the unique phenotypic signatures of human microglia, in comparison to the circulating immune cells[22]. Whether these phenotypic and functional diversities in different body compartments are more pronounced in neurological disorders or associated with any soluble mediators (i.e. biomarker) remain to be investigated.

In this study, we employed a combination of multiple state-of-the-art technologies such as high-dimensional mass cytometry (CyTOF), Seahorse, Luminex and tandem mass spectrometry to comprehensively characterize immune cells (with a particular emphasis on myeloid cells) in different body compartments including the peripheral blood, CSF, brain parenchyma (frontal cortex) and choroid plexus (CP). Specifically, we compared the cellular composition, phenotypes and metabolic responses of myeloid cells from AD patients with control individuals and patients with other neurodegenerative or neuropsychiatric disorders. Our findings showed differences in marker expression and phenotypes of myeloid cells between body compartments, associated with an activation of the immune response including changes in cytokine/chemokine expression and cell metabolism, as well as responses to acute inflammation. In line with the previously published studies[23,24], we did not detect gross phenotypic difference between diseases within each compartment but the disease-associated phenotypic differences became more pronounced after in vitro environment and stimulation. Some of these differences were found enhanced in AD patients. Results obtained from our studies demonstrate different compositions of myeloid cells between different compartments, and suggest that responses of this cell population in AD might be related to the more compartmentalized inflammation in and/or towards the CNS. Creating such a bird´s-eye view of phenotypic and functional changes of myeloid cells and other immune cells in different compartments will aid better understanding of the pathophysiology of AD at the systems-level, and potentially help improve the diagnosis and treatment of the disease.

## Results

### Differential phenotypes between peripheral blood and CSF myeloid cells

CSF and peripheral blood samples allow a more precise diagnosis as well as follow-up analyses on the same patients to monitor disease progression and/or treatment efficacy. Therefore, general knowledge of immune cell heterogeneity as well as inflammatory mediators detected in these two compartments is important for evaluating disease-associated alterations. To compare the different phenotypes of circulating immune cells in the CSF and peripheral blood, we simultaneously profiled peripheral blood mononuclear cells (PBMCs) and immune cells from the CSF that were isolated from the same donors with no neurological disorders (referred to as a control group, CON) and from patients with neurodegenerative disorders (i.e. AD, mild cognitive impairment (MCI) and Huntington's disease (HD)) (Supplementary Table 1). The immune phenotypes were characterized using our previously validated CyTOF workflow with some optimization (see Methods for more details)[22]. Briefly, the samples were first stained with an antibody panel (35 antibodies, *Panel 1*; Supplementary Table 2), which was designed to encompass the major circulating immune cell subsets (i.e. T & B cells, myeloid cells (i.e. monocytes, macrophages and dendritic cells), natural killer (NK) cells, activity-related markers, chemokine receptors and cell subset markers. After CyTOF acquisition, the data were pre-processed as previously described, including the steps of de-barcoding, compensation, and quality control (Supplementary Fig. 1a)[22,25–27]. The multi-dimensional scaling (MDS) plot[25] showed overall differential marker expression between CSF cells and PBMCs (Fig. 1a). This phenotypic variance may mainly be explained by differential expression of CD3, CD14, MRP14, CD8a, CD4, CD61, CD11c, CD35, CD38 and CCR5 as shown by the MDS-based non-redundancy score (NRS)[25] of each sample (Fig. 1b). Differences in cell compositions between CSF and blood can be illustrated in the UMAP plot (Fig. 1c). To further evaluate the phenotypic differences of immune cells between the two compartments, we performed the clustering analysis using the *FlowSOM*[28] and *ConsensusClusterPlus*[29] algorithms. To achieve a robust phenotypic differentiation between the single cells, we selected lineage markers and the top ten highest NRS markers (Fig. 1b) as input (i.e. embedding markers) for the clustering analysis. These markers (here referred to as "TYPE" markers) mainly determined phenotypic differences between the cell clusters. The rest of the markers were left as "STATE" markers, which were then used to analyze differential marker expression of each cluster between conditions.

A total of twenty clusters were identified (Fig. 1d, e and Supplementary Fig. 1b, c). Comparing the peripheral blood with the CSF compartment, we detected fifteen differentially abundant clusters between the two compartments, including myeloid cells (Clusters 4, 9, 13, 15 and 16) and NK cells (Clusters 2, 3, 8 and 20) as well as T cells (Clusters 7, 14, 17 and 19) and B cells (Cluster 5 and 6) (Fig. 1f). The CSF-enriched clusters were mainly identified as myeloid cells (Clusters 13 and 15). In addition, the proportion of CD8[+] and CD4[+] T cells in the CSF was also found higher than in the peripheral blood (Clusters 7, 17 and 19) (Fig. 1f and Source Data). Compared to classical monocytes (Cluster 16), CSF-enriched myeloid cells (Cluster 13 and 15) showed different expression of markers involved in inflammatory responses, phagocytosis and metabolism such as increased CD16, CCR5, CXCR3, CD115, CD74, GPR56, C3, ApoE and CD61 expression, whereas CD38, IL-6, TNF, CD35, CD369 (Clec7A), CD14, EMR1 and MRP14 were detected at a lower level (Fig. 1g; Supplementary Fig. 1d and Source Data). However, we detected no differences in myeloid cell heterogeneity, when comparing myeloid cells in CSF or peripheral blood between conditions

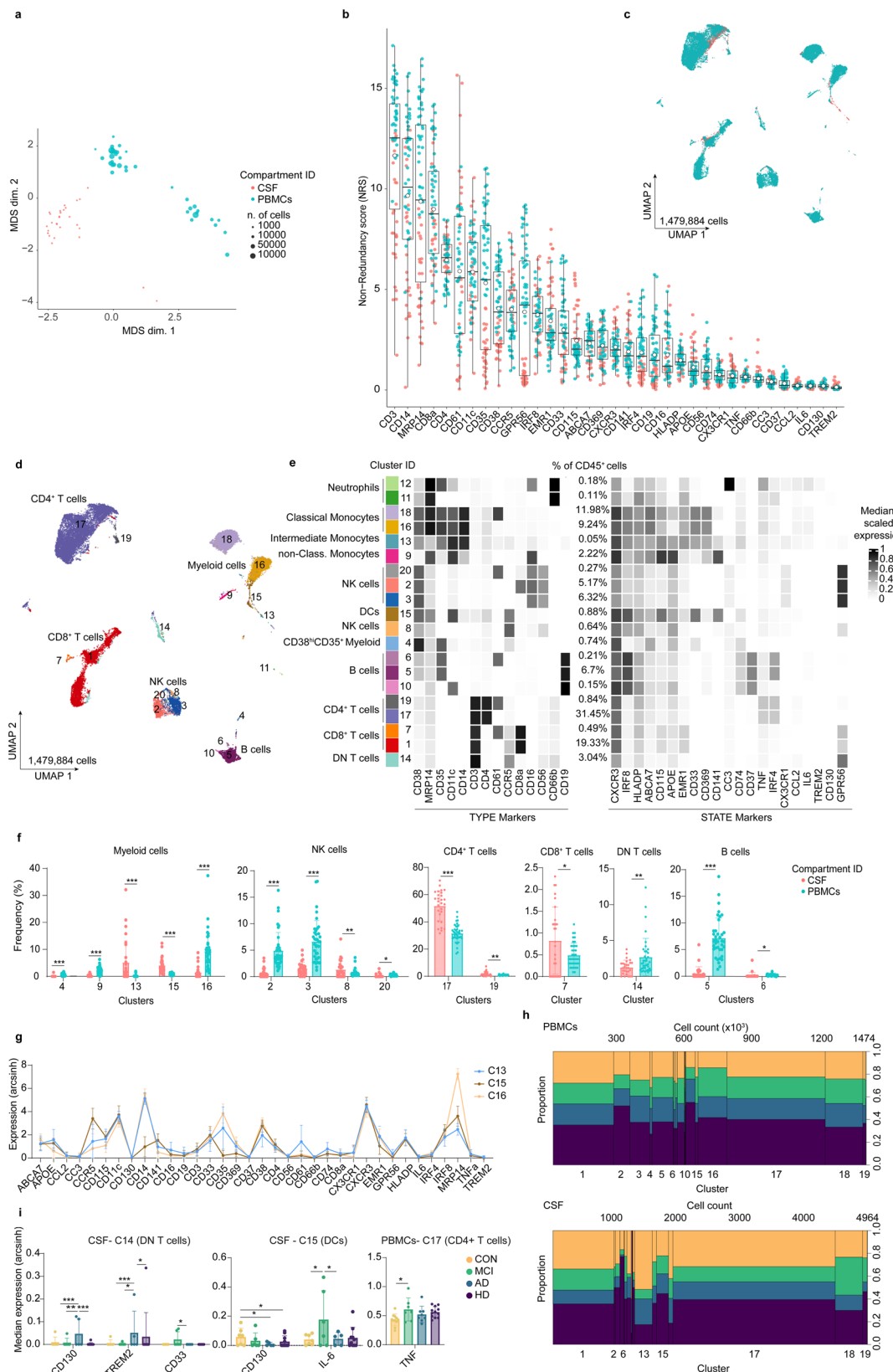

(i.e. CON, MCI, AD and HD) (Fig. 1h). Small phenotypic differences in CSF myeloid cells (Cluster 15) were detected between conditions (Fig. 1i). In addition, we also observed increased expression of TNF in blood CD4+ cells (Cluster 17) of patients with MCI, AD and HD, as well as increased CD130, TREM2 and CD33 expression in CSF double-negative (DN) T cells (Cluster 14) of AD and MCI patients (Fig. 1i).

## Increased proportion of P2Y₁₂-expressing myeloid cells within the CSF compartment

Recently, the studies[30,31] using single-cell RNA sequencing (scRNA-Seq) reported a rare population of myeloid cells in the CSF with a transcriptomic signature matching microglia. These microglia-like cells were proposed to be found only in the CSF of subjects with

**Fig. 1 | Deep immune profiling of human mononuclear cells from blood (PBMCs) and cerebrospinal fluid (CSF)−Panel 1.** Results were from 38 biologically independent PBMCs (control (CON), $n = 11$; Alzheimer's disease (AD), $n = 8$; Mild cognitive Impairment (MCI), $n = 7$ and Huntington's disease (HD), $n = 12$) and 28 biologically independent CSF samples (CON, $n = 7$; AD, $n = 5$; MCI, $n = 6$ and HD, $n = 10$). **a** MDS plot for PBMC (green dots) and CSF (pink dots) samples. **b** The plot shows NRS for each marker. Each dot represents the per-sample NR scores. Boxes extend from the 25th to 75th percentiles. Whisker plots show the min (smallest) and max (largest) values. The line in the box denotes the median. The empty black circles are mean NR scores. **c** UMAP projections of CD45$^+$ cells from PBMCs and CSF samples, coloured by compartment ID. Each dot represents one cell. **d** UMAP projection, colouring indicates 1−20 clusters. **e** Heatmap cluster depicting the median expression levels of TYPE and STATE markers. Heat colours of expression levels have been scaled for each marker individually (to the 1st and 5th quintiles)

(black, high expression; white, no expression). **f** Frequency plots of the fifteen differentially abundant clusters (mean ± SD). An FDR-adjusted $p$-value < 0.05 was considered statistically significant, determined using the EdgeR test for differential cluster abundance included in the *diffcyt* package for R (*$p < 0.05$; **$p < 0.01$; ***$p < 0.001$, adjusted). Each dot represents the value of each sample. Data displayed as mean ± SD. **g** Line graph of the arcsinh marker expression (mean ± SD) between CSF-enriched myeloid cell clusters (C13 and C15) and the classical monocytes (C16) (FDR-adjusted Mann−Whitney $U$-test, two-sided, adjusted, two-sided). **h** Mosaic plots depicting cluster proportion and cell count per cluster. **i** Median expression (with arcsinh transformation) of markers found differentially expressed between conditions using LMM (linear mixed-models) included in the *diffcyt* package for R; *$p < 0.05$; **$p < 0.01$; ***$p < 0.001$, adjusted. Data displayed as mean ± SD. Raw data (for **f**, **g** and **i**) are provided as Source Data.

neuroinflammation. However, due to the limited number of cases and the lack of proper controls, conclusions about this neuroinflammation-restricted microglia-like cells in the CSF cannot readily be drawn from these datasets. To prove the existence of this rare cell population, we performed another CyTOF measurement of CSF cells and PBMCs from the same individuals (as in Fig. 1) using an antibody panel including microglia markers such as the P2Y$_{12}$ receptor and markers involved in cell activation (*Panel 2*, 35 antibodies, Supplementary Table 3). Similar to results obtained from the antibody *Panel 1* (Fig. 1), CSF cells and PBMCs were phenotypically different, as shown in the UMAP illustration (Fig. 2a) and MDS plot (Supplementary Fig. 2a). The clustering analysis using lineage markers and the top ten NRS markers (Supplementary Fig. 2b) including P2Y$_{12}$ as embedding parameters revealed twenty clusters (Fig. 2b, c and Supplementary Fig. 2c, d). With this antibody *Panel 2*, we detected thirteen differentially abundant clusters between the two compartments, including myeloid (Clusters 4, 8, 13, 16, 17, 18 and 20), lymphoid (Clusters 1, 2, 3, 11 and 14) and NK (Cluster 5) cell subsets (Fig. 2d and Source Data). In line with the previous studies with scRNA-Seq[30,31], we detected a strong difference in the abundance of CCR2$^{low}$P2Y$_{12}^+$ (Cluster 16) and CCR2$^+$P2Y$_{12}^{lo}$ (Cluster 20) myeloid cell subsets, which were enriched in the CSF (Fig. 2d, e). However, these myeloid cell subsets were not restricted to the CSF as has been previously proposed[30,31]. CCR2$^{low}$P2Y$_{12}^+$ myeloid cells could be found also in the peripheral blood at a lower proportion (Cluster 16; *mean ± sd*: CSF, 4.95 ± 6.08, PBMCs, 0.40 ± 0.38). Our findings suggest that these cells should be cautiously termed "neuroinflammation-associated microglia-like cells", as these cells were also present in the CSF of healthy donors. When compared to the CD16$^-$CCR2$^{hi}$ classical blood monocytes (cluster 15), this CSF-enriched CCR2$^{low}$P2Y$_{12}^+$ cluster (Cluster 16) showed differences in marker expression including higher level of CD91, CD11c, HLA-DR, CD16, CD68, MS4A4A, and AXL but lower level of OPN, CCR2, CD163 and CD64 (Fig. 2f and Source Data). However, when compared with CD16$^+$CCR2$^{low}$ non classical (cluster 13) P2Y$_{12}^{lo/-}$ blood monocytes, cluster 16 show higher expression of most of the markers including HLA-DR, CD16, CCR2, CD11b, CD169, CD64, CD91, CD68 and MIP-1β (Fig. 2f and Source Data). We also observed differential marker expression between the two P2Y$_{12}^{low/+}$, CSF-enriched myeloid cell subsets (Cluster 16 and 20; Fig. 2g and Source Data), suggesting two different myeloid cell subsets/states.

Similar to the results obtained from *Panel 1*, we did not detect any major changes in cell composition between diseases and also between CON and disease CSF cells (Fig. 2h). However, some differential marker expressions (mainly within the myeloid cell subsets) could be found between CON and HD PBMCs (Fig. 2i). Of note, although it was not significantly different, similar changes in phenotypes were found in PBMCs from AD and MCI, when compared to CON PBMCs (Fig. 2i).

### Changes in glucose metabolism of CSF-treated myeloid cells

Differences in metabolic profiles can link to functional changes of myeloid cells[32]. We next evaluated changes in cellular bioenergetics of

myeloid cells after exposure to CSF. Myeloid cells from a healthy individual were isolated from PBMCs using magnetic activated cell sorting (MACS), and subsequently treated with CSF of CON individuals or of patients with AD or MCI. Bioenergetics was measured by Seahorse. Shortly after exposure to CSF, myeloid cells increased the extracellular acidification rates (ECARs), whereas plasma-treated cells showed comparable ECARs between conditions including the PBS-treated cells. The oxygen consumption rate (OCR) was slowly changed after treatment with CSF (Fig. 3a). Interestingly, myeloid cells treated with CSF of patients with AD showed slightly but significantly higher ECARs when compared with cells treated with PBS or CSF from CON individuals. ECARs were comparable between the treatment with AD-CSF and MCI-CSF. The OCR was found comparable between disease conditions. When the cells were treated with plasma, no significant differences were observed between the groups. These results demonstrate changes in metabolism (possibly in glycolysis) of myeloid cells after exposure to the CSF environment, which was enhanced in AD. Next, to precisely evaluate the alteration of glucose metabolism, we cultured pre-sorted monocytes (from the same healthy individual as in the Seahorse experiment) in the presence of 1,2-$^{13}$C$_2$-glucose and CSF from CON or AD patients, with or without LPS. $^{13}$C$_2$-glucose-derived metabolites were then quantified using HPLC-MS/MS (Fig. 3b). Similar to the findings mentioned above, we observed significantly increased glycolysis in monocytes treated with AD-CSF, determined by an increase of $^{13}$C$_2$-pyruvate production (Fig. 3c, d). However, the conversion of $^{13}$C$_2$-pyruvate to $^{13}$C-lactate was significantly decreased in AD-CSF-treated monocytes, whereas no differences were found in CON-CSF in comparison to untreated monocytes (Fig. 3c, e). Furthermore, we also detected decreased conversion of $^{13}$C$_2$-glucose to $^{13}$C$_2$-serine in AD-CSF-treated monocytes (Fig. 3c, f). These findings suggested an increased glycolysis in myeloid cells after exposure to CSF (compared with no CSF treatments), possibly resulting in an increased level of metabolites downstream of pyruvate metabolism.

### Changes in phenotypes and immune responses of myeloid cells after exposure to cerebrospinal fluid

Next, we investigated whether these metabolic changes relate to CSF-enhanced activation phenotypes of myeloid cells. A comparative measurement of pro- and anti-inflammatory mediators in the plasma and CSF from patients with AD, compared with healthy control and patients with other neurological disorders such as MCI, HD, as well as frontotemporal lobar degeneration (FTLD), depression and schizophrenia (SCZ). The Luminex assay targeted 12 proteins including IL-8, IL-6, IL-10, CCL2, TNF, as well as IP-10 (CXCL10) and Macrophage Inflammatory Protein (MIP)-α and -β, the chemokine receptor ligands of CXCR3 and CCR5 respectively. Of note, both CXCR3 and CCR5 were found up-regulated in the CSF myeloid cells (as shown in Fig. 1). The Luminex assay revealed a higher concentration of IL-8, MIP-α, MIP-β, CCL2 (MCP-1), IL-6 and IP-10 in the CSF, whereas the level of the TNF and Rantes (CCL5) were higher in plasma (Fig. 4a).

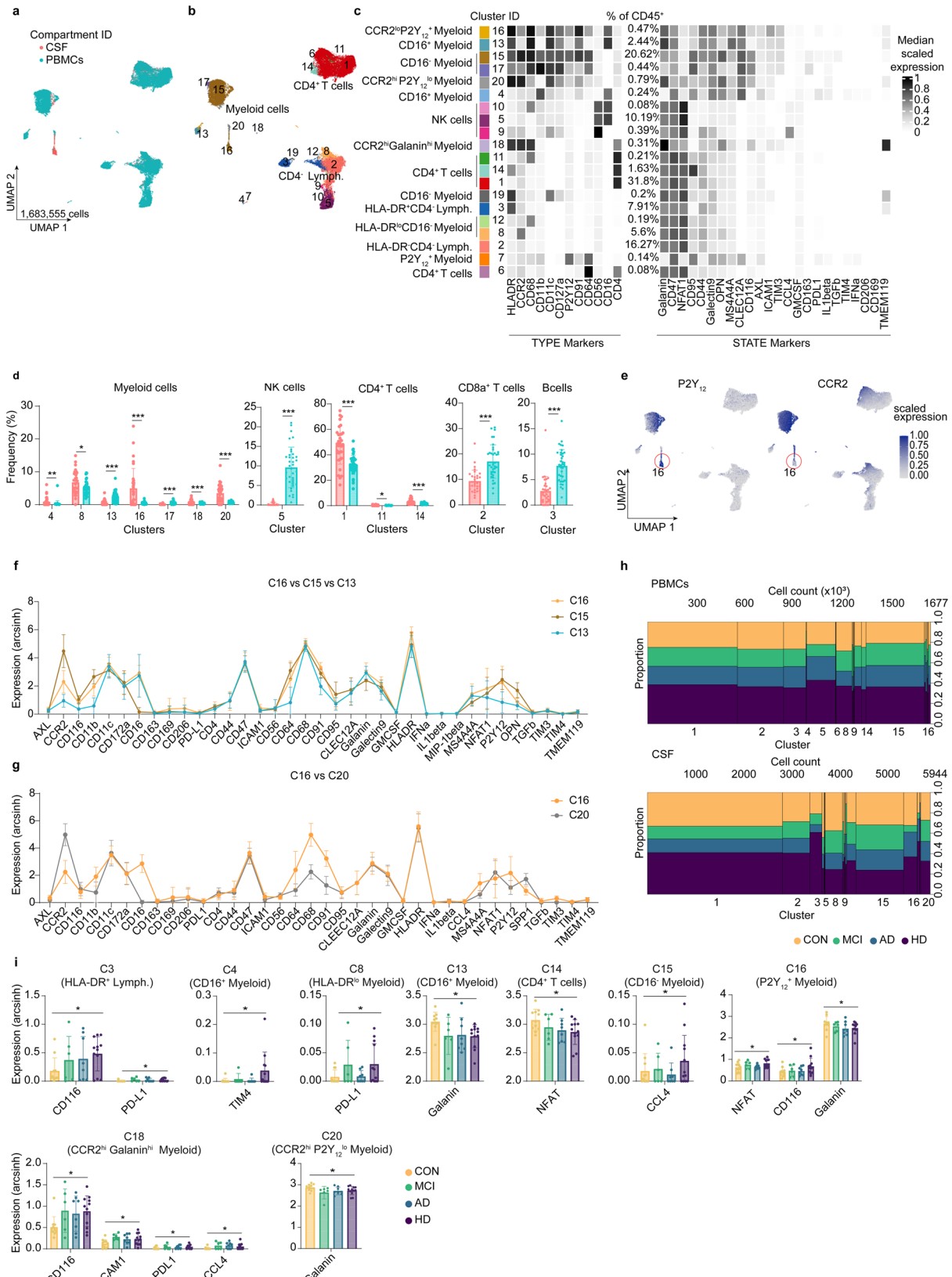

Interestingly, a potent chemotactic factor for myeloid cells, IL-8 (or CXCL8) and a CCR5-ligand MIP-α were both detected at a higher level in AD-CSF, compared to the CON- and HD-CSF, which is well correlated with the finding described above (i.e. Fig. 1) of an increased CCR5 expression on CSF-enriched myeloid cells. A previous study has demonstrated that, after exposure to plasma from patients with AD,

the human monocytic cell line THP-1 increased glycolysis and the expression of inflammatory molecules such as IL-8 and TNF[33]. Of note, IL-8 level in AD-CSF was also slightly higher than that in MCI-CSF, whereas MIP-α levels were comparable between AD- and MCI-CSF (Fig. 4b). The plasma concentration of all mediators was not significantly different between the conditions. Increased IL-8 level in

**Fig. 2 | Deep immune profiling of human mononuclear cells from blood (PBMCs) and CSF—*Panel 2*.** Results shown in **a**–**i** were obtained from 38 biologically independent PBMCs (CON, $n = 11$; AD, $n = 8$; MCI, $n = 7$ and HD, $n = 12$) and 28 biologically independent CSF samples (CON, $n = 7$; AD, $n = 5$; MCI, $n = 6$ and HD, $n = 10$). **a** Two-dimensional projections of single-cell data generated by UMAP of PBMCs (green dots) and CSF (pink dots) cells. Each dot represents one cell. **b** UMAP plot of all samples. The colouring indicates 20 clusters representing diverse immune cell phenotypes, defined by the *FlowSOM* algorithm. **c** Phenotypic heatmap of cluster identities depicting the expression levels of 12 TYPE markers used for the cluster analysis and 24 STATE markers. Heat colours of expression levels have been scaled for each marker individually (to the 1st and 5th quintiles) (black, high expression; white, no expression). **d** Frequency plots of the thirteen differentially abundant clusters between the CSF and blood compartments. Data

displayed as mean ± SD. An FDR-adjusted *p*-value < 0.05 was considered statistically significant, determined using the edgeR test for differential cluster abundance (**p* < 0.05; ***p* < 0.01; ****p* < 0.001, adjusted). Each dot represents the value of each sample. **e** Overlaid UMAP plots of all samples showing scaled expression of P2Y$_{12}$ receptor and CCR2. **f**, **g** Line graph of the arcsinh marker expression (mean ± SD) between CSF-enriched myeloid cell cluster (C16) and the classical monocytes (C15) (**f**) and CCR2$^+$P2Y$_{12}$$^{lo}$ (Cluster 20) (**g**) (FDR-adjusted Mann–Whitney *U*-test, two-sided, adjusted, two-sided). **h** Mosaic plots depicting cluster proportion and cell count per cluster for PBMC and CSF cells. **i** Median expression (with arcsinh transformation) of markers found differentially expressed between conditions (CON, MCI, AD and HD) using LMM (linear mixed-models) included in the diffcyt package for R; **p* < 0.05; ***p* < 0.01; ****p* < 0.001, adjusted. Data displayed as mean ± SD. Source data (for **d**, **f**, **g** and **I**) are provided as a Source Data file.

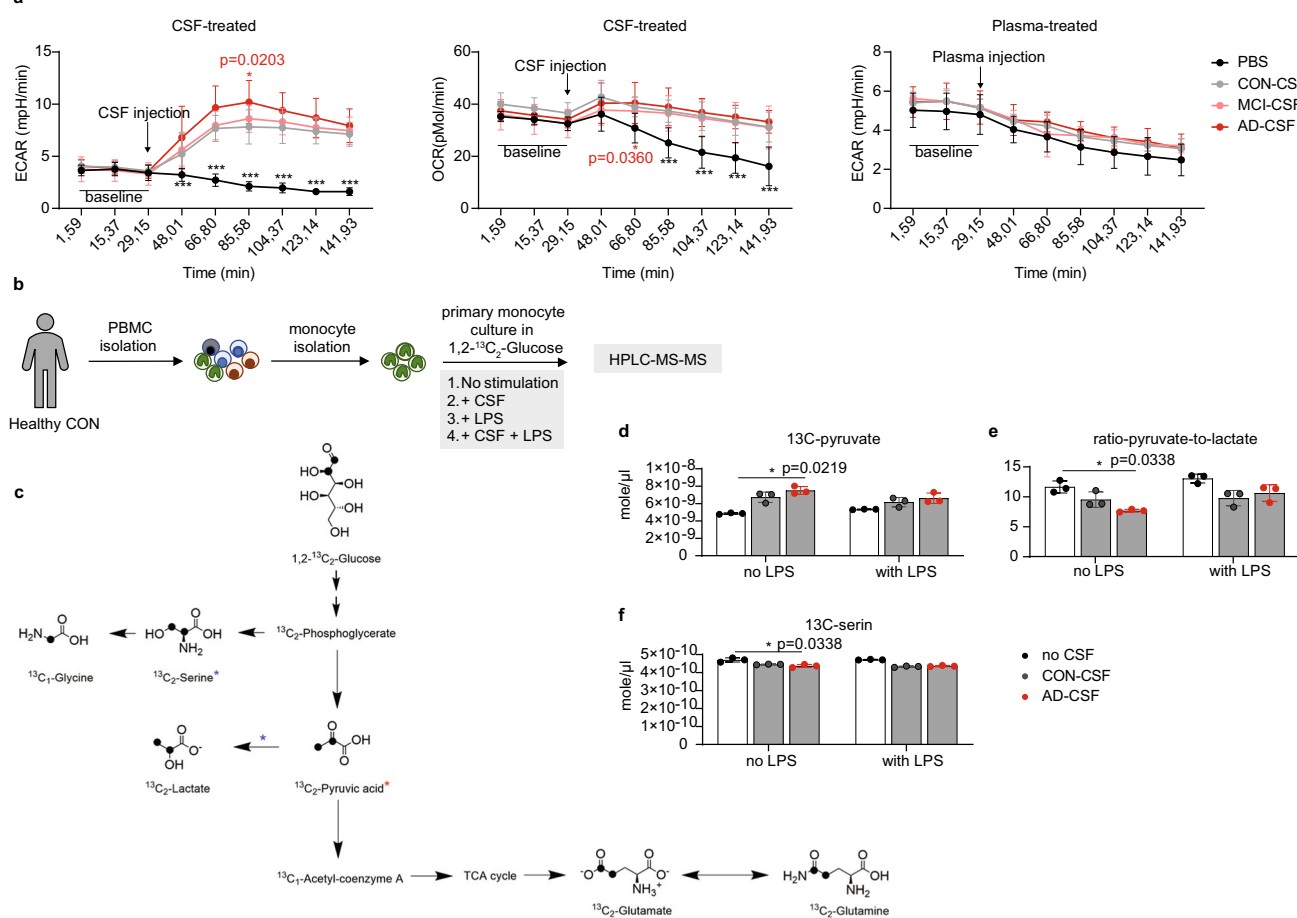

**Fig. 3 | Alterations in glucose metabolism after exposure to CSF. a** Seahorse analysis of extracellular acidification rate (ECAR, right and left plot) and oxygen consumption rate (OCR, middle plot) in monocytes treated with either PBS, CSF or plasma from CON ($n = 5$, biological replication), MCI ($n = 5$, biological replication) or AD ($n = 5$, biological replication). Data displayed as mean ± SD. Two-way ANOVA with Tukey's multiple comparisons; **p* < 0.05, ***p* < 0.01, ****p* < 0.001 and *****p* < 0.0001. The results are from three different experiments. **b** Schematic representation of sample processing, ex vivo experiment and measurement. Reference PBMCs from a healthy donor were isolated, monocytes were pre-sorted using the magnetic activated cell separation (MACS). Cells were then cultured in the

presence of 1,2-$^{13}$C$_2$-glucose and treated with different conditions. Cells were then harvested and analyzed with HPLC-MS-MS. (**c**–**f**) The 1,2-$^{13}$C$_2$-glucose metabolism (**c**) showing metabolites obtained from monocytes treated either with PBS or CSF from CON ($n = 3$, biological replication) or AD ($n = 3$, biological replication) that were quantified using HPLC-MS-MS. Red asterisk indicates significantly increased metabolite, whereas the blue asterisk labels significantly decreased metabolites. The bar graphs show concentration of 1,2-$^{13}$C$_2$-glucose-derived $^{13}$C-pyruvate (**d**), the $^{13}$C-pyruvate-to-$^{13}$C-lactate conversion (**e**) or $^{13}$C$_2$-glucose-derived $^{13}$C-serine (**f**). Data displayed as mean ± SD. Kruskal–Wallis test, **p* < 0.05 and ***p* < 0.01. Source data (for **a**, **d**, **e** and **f**) are provided as a Source Data file.

AD-CSF was positively correlated with IL-6 and MIP-α (Fig. 4c), suggesting that the CSF-conditions potentially facilitated changes in myeloid cell phenotypes towards the CNS. Of note, the level of the cytokine expression did not correlate with the age in patients with AD and MCI (Fig. 4d). We detected gender-related differences in the CSF IL-8 level in CON (higher level in males) and MCI (higher level in females) samples (Supplementary Fig. 3a). The level of CSF MIP-1α

was comparable between genders in all conditions (Supplementary Fig. 3b).

To determine the impact of IL-8 or MIP-1α on phenotypic changes of blood immune cells, PBMCs isolated from CON donors and patients with AD and MCI were cultured in the presence of either paired CSF (from the same individual) or IL-8 or MIP-α, using our previously validated protocol[34]. PBS-treated culture served as a control for the

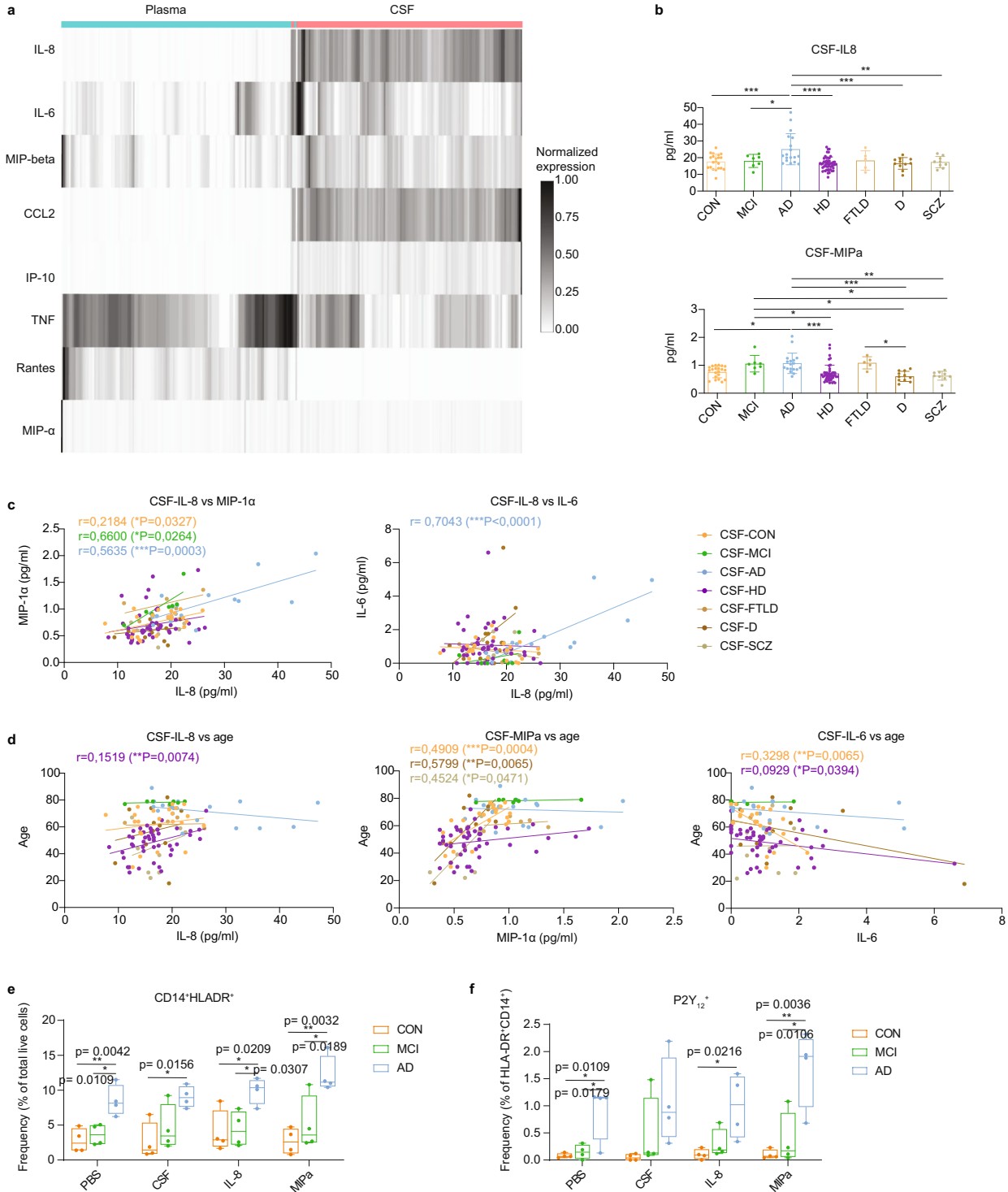

**Fig. 4 | Differential concentration of inflammatory mediators between plasma (n = 117, biological replication) and CSF (n = 117, biological replication) determined using Luminex protein array.** These are CON individuals (n = 21) and patients with MCI (n = 7), AD (n = 18), HD (n = 46), depression (n = 11), FTLD (n = 5) and SCZ (n = 9) individuals. **a** Heatmap showed the expression of IL-8, IL-6, MIP-1β, CCL2, IP-10, TNF, Rantes (CCL5) and MIP-1α that were quantified in plasma and CSF. **b** Bar graphs show differential concentrations (pg/ml) of IL-8 and MIP-α in the CSF of patients with AD, compared to CON individuals and patients with other diseases. Each dot represents the value of each sample. Data displayed as mean ± SD. One-way ANOVA with the Tukey's multiple comparisons; *p < 0.05, **p < 0.01 and ***p < 0.001. **c** Scatter plots showing correlation between mean CSF-concentration of IL-8 and MIP-α or IL-6 of all groups. *P < 0.05, **P < 0.01 and ***P < 0.001, nonparametric Spearman correlation test (r), two-sided. **d** Scatter plots showing correlation between mean IL-8 and MIP-1α or IL-6 concentration and age of all groups. *P < 0.05, **P < 0.01 and ***P < 0.001, nonparametric Spearman correlation test (r), two-sided. **e, f** Results obtained from in vitro experiments, in which PBMCs from CON (n = 4, biological replications), AD (n = 4, biological replication) and MCI (n = 4, biological replication) individuals treated with either PBS, CSF, IL-8 or MIP-1α. Flow cytometry analysis revealed increased frequency of CD14+HLA-DR+ cells (% of total live cells, **e**) and P2Y12+ cells (% of HLA-DR+CD14+ cells, **f**) in AD-PBMCs after in vitro incubation (PBS) or after treatments (CSF, IL-8 and MIP-1α). Boxes extend from the 25th to 75th percentiles. Whisker plots show the min (smallest) and max (largest) values. The line in the box denotes the median. Each dot represents the value of each sample. Ordinary one-way ANOVA; *p < 0.05, **p < 0.01, ***p < 0.001. Source data (for **b**, **c**, **d**, **e** and **f**) are provided as a Source Data file.

in vitro environment. Flow cytometry analysis (Supplementary Fig. 4) revealed higher proportion of the CD14$^+$HLADR$^+$ myeloid cell subset in AD-PBMCs in vitro, compared to CON- and MCI-PBMCs. However, this difference was comparable between AD-PBMCs treated with CSF, IL-8, MIP-1α and PBS (Fig. 4e), suggesting that AD-PBMCs are more vulnerable to environment changes such as transferring cells to an in vitro environment or possibly from the peripheral blood to the CSF. Interestingly, this CD14$^+$HLA-DR$^+$ cell population in AD showed higher proportion of P2Y$_{12}$$^+$ cells, compared to CON and MCI, but the proportion remained comparable between treatments (Fig. 4f). To in-depth characterize phenotypic and functional changes of myeloid cells after exposure to CSF, we performed another in vitro experiment using PBMCs isolated from CON individuals, and patients with AD and MCI. The PBMCs were treated with either only paired-CSF or both paired CSF and lipopolysaccharide (LPS), and labelled with a CyTOF-antibody panel (see Supplementary Table 4 for the panel of 37 antibodies). The CD3$^-$CD19$^-$ cell population was first pre-gated and analyzed using the data analysis workflow as described above (Figs. 1, 2 and Supplementary Fig. 1a). The MDS plots showed no completely clear phenotypic differences between conditions (i.e. no stimulation, CSF and CSF + LPS) or between diseases (i.e. CON, MCI and AD) (Fig. 5a, b). Using clustering analysis, 20 clusters were identified (Fig. 5c, d). Similar to results obtained from flow cytometry analysis, we detected increased proportion of CD14$^+$HLADR$^+$ cells (cluster 1) in AD-PBMCs, compared to the CON-PBMCs, especially in AD-PBMCs treated with CSF and LPS (Fig. 5e). These cells are characterized as CD14$^+$HLADR$^+$CD16$^+$CCR2$^{low}$CD68$^{hi}$CD11c$^+$CD141$^+$CCR6$^+$ (Fig. 5d), which have a similar phenotype as the CCR2$^{low}$CD68$^{hi}$CD11c$^+$HLADR$^+$D16$^+$ CSF-enriched myeloid cells (cluster 16 in Fig. 2). Interestingly, the expression of P2Y$_{12}$ receptor, IL-8, MIP-1β,TNF and CXCR3 in this cluster were found to be increased after exposure to CSF and after treatment with LPS, especially in PBMCs from patients with AD (Fig. 5f). These findings support our results shown in Figs. 1–2 (i.e. increased activation phenotype of myeloid cells after exposure to CSF). The results also strengthen our hypothesis that the P2Y$_{12}$$^+$ myeloid cells detected in CSF may be derived from the peripheral blood myeloid cells (possibly from CD14$^+$CD16$^+$CCR2$^{low}$ monocytes), which increase P2Y$_{12}$ expression upon entry to the CSF compartment. Furthermore, our findings demonstrated that AD-PBMCs are more vulnerable than CON- or MCI-PBMCs to inflammatory stimulation. In addition, we also detected an increased abundance of CD56$^+$CD11c$^+$ NK cells (cluster 4, Fig. 5g and Source Data) in MCI-PBMCs, compared with CON- and AD-PBMCs (Fig. 5h). But the proportion of this population was comparable between treatments (i.e. no stimulation, CSF and CSF + LPS). However, we detected increased expression of CXCR3 of CD56$^+$CD11c$^+$ cells after exposure to CSF and CSF + LPS (Fig. 5i), suggesting phenotypic changes also in the NK cell population.

Next, we treated PBMCs isolated from a healthy donor with CSF obtained from six control individuals or six patients with AD. In addition to CSF, PBMCs were also stimulated with LPS (similar to the experiment shown in Fig. 5). Cells were analyzed using CyTOF work-flow as described above. Twenty clusters were identified (Fig. 6a; Supplementary Table 5 for the panel of 40 antibodies used in Fig. 6). An MDS plot showed small changes in phenotype of myeloid cells after co-incubation with CSF, compared to non-stimulated cells (Fig. 6b). As expected, cells treated with LPS and both CSF and LPS (CSF + LPS) showed distinct overall phenotypes, compared with no stimulation and CSF-treated groups (Fig. 6b). We detected increased proportions of two small clusters with mixed phenotypes (cluster 6 and 15) in CSF and CSF + LPS-treated PBMCs (Fig. 6c, d). Compared with all other cell subsets, these two clusters expressed higher level of CD69, CD49d and IL-10 (Fig. 6e). Similar to CD14$^+$CD16$^+$CCR2$^{low}$ CSF-enriched myeloid cell subsets described above (Figs. 2 and 5), CD14$^+$ cluster 15 showed different phenotypes when compared to the CD14$^+$CCR2$^+$ monocytes (Cluster 19). These differences include a higher expression level of

P2Y$_{12}$ and CD16, but a lower CCR2 expression (Fig. 6f, g). In addition, myeloid cell subsets (Cluster 8, 16, 18 and 19) in CSF + LPS-treated samples showed consistently higher expression of inflammatory cytokines such as TNF, MIP1-β and IL-1β (Fig. 6h), in comparison to LPS-treated PBMCs. Of note, no differences in phenotypes and responses to LPS were detected between cells treated with AD-CSF and those treated with CON-CSF.

Together, we detected big changes in metabolic profiles of mye-loid cells after exposure to CSF in vitro (as shown in Fig. 3), but only small changes in phenotypes and responses to LPS of these cells could be detected after CSF treatment (as shown in Figs. 5 and 6). Incubating myeloid cells from patients with AD in the presence of the paired CSF (with and without LPS) could induce more phenotypic changes than when healthy myeloid cells were treated with CSF (with and without LPS). Our findings suggest that myeloid cells of patients with AD may already be primed in the peripheral blood (possibly without significant changes in phenotypes), and thus are more vulnerable to LPS stimulation.

## Differential myeloid cell phenotypes in choroid plexus (CP) and brain parenchyma

Next, we characterized differences in phenotypes and abundant clus-ters of myeloid cells between the brain barrier (i.e. choroid plexus, CP) and the brain parenchyma (i.e. gyrus frontalis medialis, GFM). Since the brain tissue of AD patients is only available post-mortem, we were not able to perform the investigation in the same individuals as were used for the blood and CSF determination. Furthermore, it should be mentioned that this study is limited to the changes at the late state of the disease (i.e. post-mortem).

We performed CyTOF analyses with pre-sorted CD45$^+$ cells iso-lated from post-mortem CP and GFM of AD and non-neurological donors (Supplementary Table 1 and Supplementary Fig. 5). First, we characterized isolated cells using an antibody panel (*Panel A*), focusing on known immune cell populations (see Supplementary Table 6 for the list of 35 antibodies used). The UMAP plots of the obtained CyTOF data showed different cellular compositions between CP and GFM com-partments (Fig. 7a). The downstream clustering analysis of all data together (i.e. across all cells of all CP and GFM samples) revealed a total of 14 distinct cell clusters (Fig. 7b, c). Higher proportion of macro-phages, monocytes and lymphocytes was detected in CP, whereas IRF8$^{hi}$ (Cluster 9, 10 and 12) microglia/macrophages were the majority in GFM (over 80% of total cells, Fig. 7d). Similar to previous study in mouse CNS[35], we detected a small population of CP-macrophages (i.e. border-associated, Kolmer´s epi-plexus macrophages, CP$^{epi}$-BAM) whose phenotypic signature reminiscent of microglia, and were clus-tered together with GFM microglia (Cluster 10: IRF8$^{hi}$CD11c$^+$HLA-DR$^+$EMR1$^+$GPR56$^+$). Compared with the GFM-microglia, CP$^{epi}$-BAM expressed a higher level of markers involved in phagocytosis and cell activation including CD206, CD64, HLA-DR, CD44, CCR5, CD68, MS4A4A, CD32 and CD14 (Fig. 7e and Source Data). Although we could not detect differences in cellular compositions within the CP com-partment between the CON and AD groups, we detected some markers differentially expressed in both myeloid and lymphoid cell populations between the groups. Collectively, we found lower expression of EMR1, IRF8, CD14, CD86 and C3 in myeloid cells from AD donors, whereas ABCA7, CD61 and CCR5 were found increased (Fig. 7f). Analogous to the CP, no differences in cluster abundance were found between AD-GFM and CON-GFM. Also in line with the results from CP analysis, myeloid cell clusters (cluster 1 and 7) in AD-GFM showed higher expression of CCR5 and CD61. The microglia cluster, cluster 12, of AD-GFM expressed lower level of CXCR3 and CD4. To further characterize myeloid cells in these two compartments, we have utilized the anti-body *Panel B* (Supplementary Table 7, a total of 35 markers) consisting of functional markers such as the thrombospondin 1 receptor CD47 (a "don´t eat me" signal, the extracellular ligand of CD172a) and MIP-1β

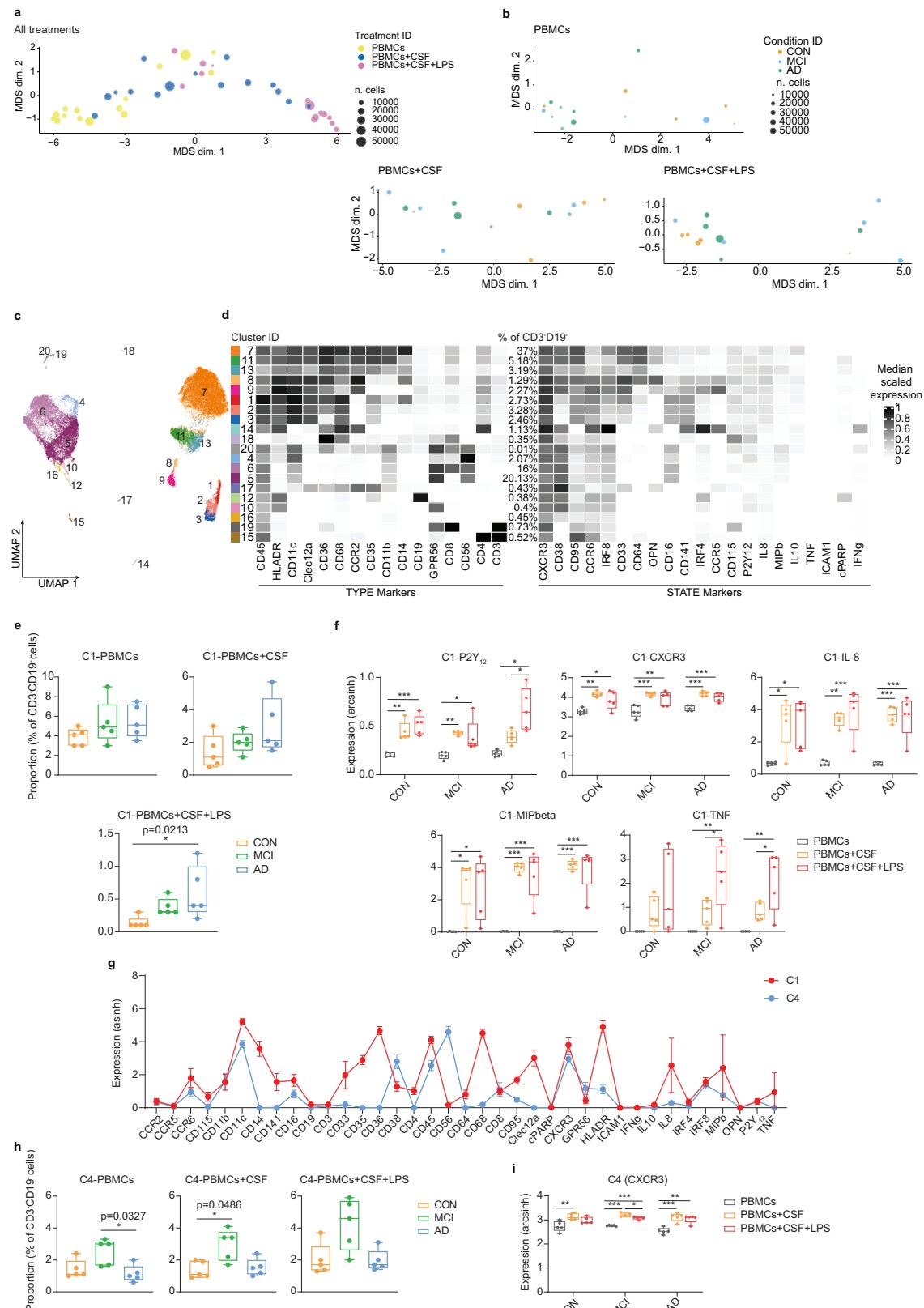

(CCL4, a CC chemokine with specificity for CCR5). Similar to the results shown in Fig. 7, the cellular composition of CP was different from that of GFM, as shown by UMAP plots (Fig. 8a). Among the 14 defined clusters (Fig. 8b, c), we detected a higher proportion of $P2Y_{12}^{low/-}$ myeloid cells in CP, compared with GFM, whereas $P2Y_{12}^{hi}$ microglia/macrophages (Cluster 8, 10 and 13) were mainly found in GFM (Fig. 8d).

The results obtained from the antibody *Panel B* confirmed the similarity between GFM microglia and the $CP^{epi}$-BAM (Cluster 8: $P2Y_{12}^{+}CD11c^{+}CD64^{+}Glut5^{+}$, Fig. 8d, e, Source Data). Similar to the *Panel A*, we could not detect differentially abundant clusters between CON and AD in both compartments using the antibody *Panel B*. Comparing AD-CP macrophages with those in CON-CP revealed increased

**Fig. 5 | Assessment of paired-CSF-induced phenotypic changes in the myeloid cells from patients with AD ($n = 5$, biological replications) and MCI ($n = 5$, biological replications), in comparison with CON ($n = 5$, biological replications) cells. a** MDS plot for all PBMCs at different conditions. **b** MDS plots for each treatment, colour-coded by condition (CON, MCI or AD). Each dot represents a sample and dot size depicts total number of cells per sample. **c** The overlaid UMAP plot of all samples. The colouring indicates 20 clusters representing diverse immune cell phenotypes, defined by the *FlowSOM* algorithm. **d** Phenotypic heat-map of cluster identities depicting the expression levels of 16 TYPE markers used for the cluster analysis and 21 STATE markers. Heat colours of expression levels have been scaled for each marker individually (to the 1st and 5th quintiles) (black, high expression; white, no expression). **e** Proportion of Cluster 1 (C1) between groups and conditions. Kruskal–Wallis test; *$P < 0.05$, **$P < 0.01$, ***$p < 0.001$ and

****$p < 0.0001$. **f** Differences in marker expression of P2Y$_{12}$, CXCR3, IL-8, MIPbeta in cells from Cluster 1 between conditions. Ordinary one-way ANOVA; *$P < 0.05$, **$P < 0.01$, ***$P < 0.001$ and ****$P < 0.0001$. **g** Line graph of the arcsinh marker expression (mean ± SD) between Cluster 1 and Cluster 4 (FDR-adjusted Mann–Whitney *U*-test, two-sided, adjusted, two-sided). **h** Proportion of C4 (% of CD3⁻CD19⁻ cells) between groups and conditions. Kruskal–Wallis test; *$P < 0.05$, **$P < 0.01$· **$p < 0.001$ and ****$p < 0.0001$. **i** Difference in marker expression of CXCR3 in cells from Cluster 4 between conditions. All boxes extend from the 25th to 75th percentiles. Whisker plots show the min (smallest) and max (largest) values. The line in the box denotes the median. Each dot represents the value of each sample. Ordinary one-way ANOVA; *$P < 0.05$, **$P < 0.01$, ***$P < 0.001$ and ****$P < 0.0001$. Source data (for **e, f, g, h** and **i**) are provided as a Source Data file.

expression of CD206, CD163, CD91, CD33, CD172a (SIRPα, the receptor of CD47) and Clec12A, whereas CD64, CD68, CD18 and CD47 were downregulated in AD-CP (Fig. 8f). These changes may reflect an inefficient or even suppressed phagocytosis in macrophages isolated from AD-CP. Both microglia subsets (cluster 10 and 13) from AD-GFM showed a higher level of the inflammatory mediator MIP-1β (CCL4, a CC chemokine with specificity for CCR5), whereas CD64 and CD172a were found downregulated in AD-microglia (Fig. 8f). The macrophage subsets (cluster 4, 6 and 9) of AD-GFM showed higher expression level of CD172a, CD11b, CD18, Clec12A and the thrombospondin 1 receptor, CD47 (Fig. 8f), suggesting increased phagocytic phenotypes of the infiltrating macrophages detected in AD-GFM, whereas microglia at this late stage of disease were rather a source of chemoattractant for monocytes and macrophages.

## Discussion

Myeloid cells including monocytes, macrophages and microglia have long been suggested as key players in neuroinflammation and neuro-degeneration like AD[6,36]. Numerous findings in rodent models of AD highlight the importance of diverse myeloid cells including microglia, monocytes and monocyte-derived macrophages[6,16] in the pathology. However, these models only partially replicate the complexity of the rare familial AD, our understanding of how myeloid cells either respond or contribute to the pathogenesis of human AD including sporadic AD is still limited. In this study, we characterized and compared myeloid cells isolated from the peripheral blood, CSF, CP and GFM of either control individuals (i.e. without neurological diseases), or patients with AD, MCI or HD. Using multiple state-of-the-art analytical methods, we identified differences in myeloid cell composition and phenotypes, as well as their bioenergetic pathway across different compartments (i.e. blood and CSF) and diseases. Our findings showed that myeloid cells alter their activation phenotypes and inflammatory responses across different compartments, and this continuum of phenotypic changes and functional responses may be more pronounced in neuropathology such as AD. Overall, we could consistently detect in the CSF an increased proportion of myeloid cells with changes in activated, inflammatory and/or phagocytic phenotypes, which was characterized by increased expression of markers involved in these processes including P2Y$_{12}$ receptor, CD16, CCR5, ApoE, CD11c, HLA-DR, CD169 (SIGLEC-1), CD91 and MS4A4A. Although differences in phenotypes and responses of myeloid cells between different conditions (i.e. CON vs diseases) were small within the peripheral blood and CSF compartment, we have detected phenotypic and metabolic changes in AD compared to the CON and/or other diseases in the CP and brain parenchyma as well as in vitro. These differences were more pronounced in AD-myeloid cells after additional stimulation with LPS, suggesting that myeloid cells from AD patients are more likely vulnerable to environment change. These results are in agreement with previous studies dismissing major phenotypic changes in unstimulated PBMCs from AD donors but that have found significantly changed upon response to stimulation[24]. On the basis of our findings,

microglia in the GFM showed less phagocytic but more inflammatory phenotypes, in comparison to the infiltrating macrophages at this late stage of the disease (i.e. post-mortem). Nonetheless, due to the limitations of our study (e.g. low cell number in the CSF, high biological variation between individuals and availability of multiple body compartments from the same individuals), the findings should be interpreted with caution.

Upon entry to the CSF compartment, myeloid cells including monocytes increased the expression of markers associated with the inflammatory process including P2Y$_{12}$ receptor, CCR5, ApoE, CD169 and its co-activator MS4A4A[37]. Interestingly, CD169 (or Siglec1) was proposed as an indicator of the activity in an inflammatory CNS, due to the results showing that CD169-expressing myeloid cells were abundantly located in an active inflammatory site of the CNS, including in active multiple sclerosis lesions, acute infectious and malignant diseases. Such cells were suggested to support the activation of adaptive immune responses[38]. Soluble triggering receptor expressed on myeloid cells 2 (sTREM2) in CSF is hypothesized to increase in response to microglial activation due to neurodegenerative processes and is elevated in AD[39,40]. MS4A4A is a key modulator of sTREM2[41]. Targeting MS4A4A at the molecular or protein level was sufficient to significantly reduce sTREM2, thus can potentially be used for AD therapy. We hypothesize that myeloid cells are recruited into the CSF and become inflammatory, which may be a mechanism to regulate the adaptive immune responses in this compartment barrier. Together with an increased expression of IL-8 (a potent chemotactic factor for myeloid cells) in the AD-CSF, which is positively correlated with IL-6 and MIP-1α (CCL3, a ligand of CCR5) expression, we propose that, in AD, CCR5-expressing myeloid cells including monocytes are recruited into the CSF, become activated and associated to the activation of adaptive immune cells. In mouse models of AD, it has been shown that Aβ could stimulate the production of IL-8 and MIP-1α from monocytes or microglia. The MIP-1α-/CCR5-signalling pathway that was induced by Aβ could result in increased lymphocyte transendothelial migration to the brain[8]. Interestingly, studies in 5xFAD mice have shown that knocking out CD33 and TREM2, both known risk factors for sporadic AD, induce changes in IL-6 and IL-8 expression by microglia, and that downregulation of both signalling molecules is associated with increased neurodegeneration[42].

In the CNS, the purinergic P2Y$_{12}$ receptor (an adenosine diphosphate responsive G protein-coupled receptor) is widely recognized as a marker that is selectively expressed on microglia. In the periphery, this receptor can however be detected in multiple cell types including eosinophils[43], platelets, osteoclasts, vascular smooth muscle cells, dendritic cells[44] and macrophages[45]. Moreover, during chronic neuroinflammation, CNS-infiltrating macrophages also acquire P2Y$_{12}$ receptor[45], thus caution should be taken in the strict definition of microglia-specific marker and plasticity of myeloid cells in the niche of different body compartments. In our previous study, we have also shown low P2Y$_{12}$ expression on brain CD206⁺ macrophage[22]. In the brain, P2Y$_{12}$-expressing macrophages/microglia were found

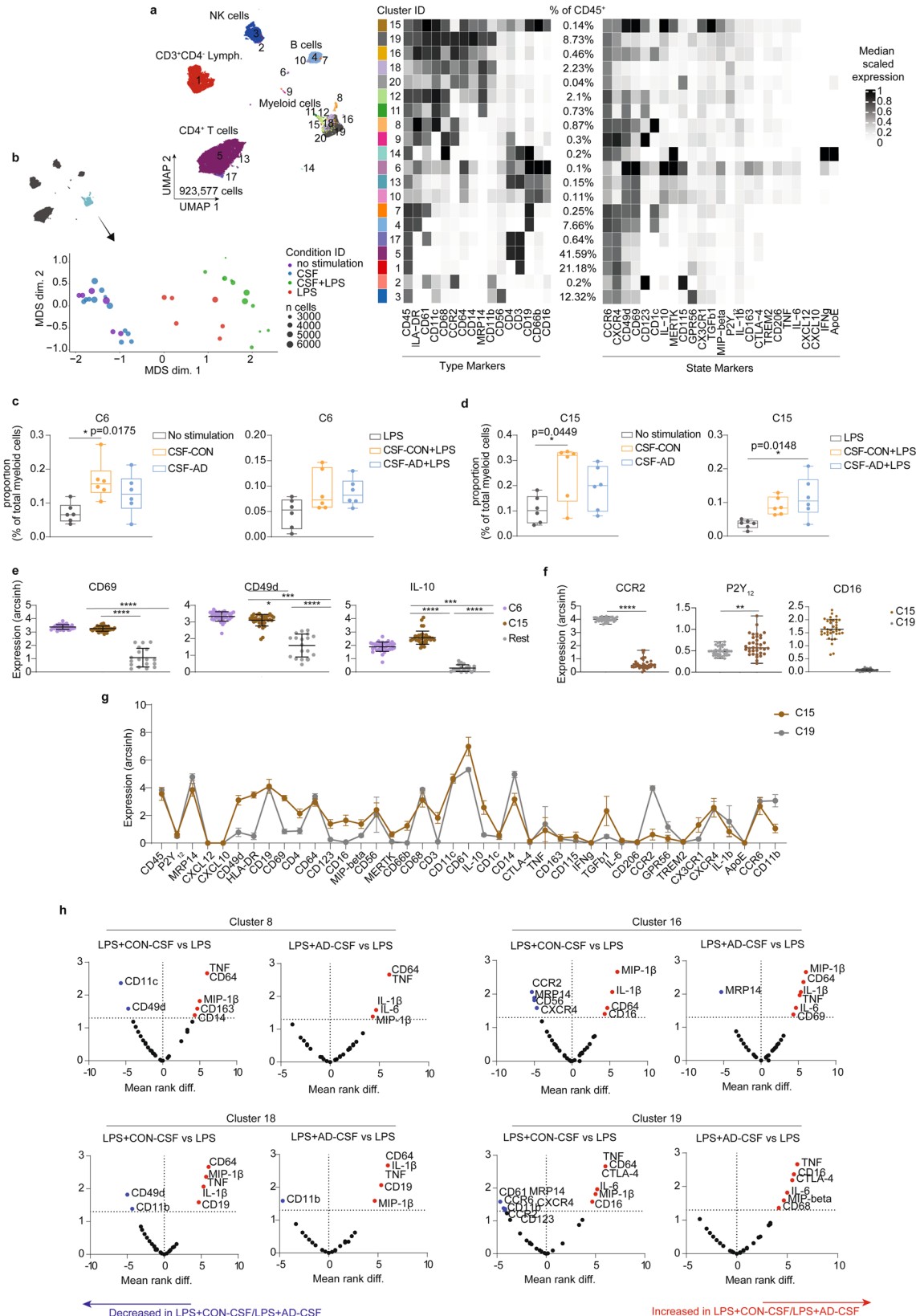

significantly reduced in multiple sclerosis and AD[26,46]. However, its role in neuroinflammation or neurodegeneration remains largely unclear. In contrast to previous studies showing that these P2Y$_{12}$$^+$ cells were microglia-like cells that are solely found in the CSF of patients with neuroinflammation[30,31], we identified these cells in both CSF (with much higher abundance) and the peripheral blood of both healthy

donors and patients with neurological disorders. Furthermore, we could also demonstrate that some subsets of blood monocytes increased P2Y$_{12}$ expression after exposure to the CSF, and thus provide an alternative hypothesis for the origin of P2Y$_{12}$$^+$ cells in the CSF, namely that they may be derived from blood monocytes that respond to (most likely) soluble factors in the new environment. Interestingly,

**Fig. 6 | Phenotypic changes in healthy PBMCs after treatment with CSF from CON ($n = 6$, biological replication) or AD ($n = 6$, biological replication) patients.** **a** UMAP plots from all cells coloured by cluster ID for 1–20 clusters determined using the *FlowSOM* algorithm, phenotypic heatmap of median marker expression per cluster. **b** MDS plot for myeloid cells (light blue dots) obtained from all conditions, i.e. no stimulation (purple dots) or CSF (blue dots), CSF + LPS (green dots) and LPS (red dots). **c**, **d** Proportion (% of CD45$^+$ cells) of Cluster 6 (**c**) or Cluster 15 (**d**) between treatments. All boxes extend from the 25th to 75th percentiles. Whisker plots show the min (smallest) and max (largest) values. The line in the box denotes the median. Each dot represents the value of each sample. Ordinary one-way ANOVA; *$P < 0.05$. **e** Mean signal intensity levels of CD69, CD49d and IL-10 staining in cluster 6 and 15, compared to the other cells (rest) (black lines show mean ± sd

values of the datasets). Kruskal–Wallis test, *$p < 0.05$ and ****$p < 0.0001$. **f** Scatter plots show the differential expression of CCR2, P2Y$_{12}$ and CD16 of Cluster 15, compared to CD14$^+$CD16$^-$ classical monocytes (Cluster 19). Data displayed as mean ± SD. Kruskal–Wallis test, **$p < 0.01$ and ****$p < 0.0001$. **g** Line graph shows different marker expressions (arcsinh) (mean ± SD) between Cluster 15 and Cluster 19. **h** Volcano plots show differential expression of all markers in myeloid cell clusters (Cluster 8, 16, 18 and 19) after AD-CSF + LPS or CON-CSF + LPS treatment, in comparison to LPS treatment. Red dots indicate markers with significantly increased expression; blue dots the markers with significantly decreased expression, whereas black dots are non-significant markers, determined using Mann–Whitney U-Test, two-sided ($p < 0.05$ is statistically significant). Source data (for **c**, **d**, **e** and **f**) are provided as a Source Data file.

treating PBMCs from AD-patients with the paired CSF in vitro resulted in increased proportion of myeloid cells showing a similar phenotype as CSF-enriched myeloid cells, characterized as P2Y$_{12}$$^+$CD14$^+$CD16$^+$C-C2$^{low}$. This change was more significant when the cells were additionally stimulated with LPS. Furthermore, we showed that this cluster produced more IL-8 and inflammatory MIP-1β after exposure to CSF and LPS, suggesting a possible source of IL-8-expressing myeloid cells in CSF.

In addition to phenotypic changes in the myeloid cells of the CSF, results obtained from Seahorse and $^{13}$C-glucose tracing experiment showed that blood monocytes are more glycolytically active after exposure to the CSF, suggesting an inflammatory response rather than anti-inflammatory phenotype, which would more likely increase mitochondrial respiration[33]. Upregulation of glycolytic metabolism was proposed to support phagocytotic function via ATP production[32].

Besides brain macrophages and microglia, myeloid cells in CP have been also proposed as one of the key players in human AD pathology. CP is a unique organ exposed to peripheral blood and CSF, forming the blood-CSF-barrier (BCSFB), which effectively separates the brain parenchyma from the peripheral blood, and regulates neuronal homoeostasis. CP allows efficient exchange of essential gases, nutrients and waste products of metabolism between blood, CSF and interstitial fluid of the brain[47]. This barrier also efficiently removes cell debris and larger waste products including β-amyloid[48]. The CP inner stroma is richly irrigated by fenestrated capillaries, which facilitate the passage of circulating macromolecules and immune cells into this compartment. However, under healthy conditions this BCSFB restricts immune cell entry into the CSF and the brain parenchyma[49]. Dysfunction of this system may play an aetiological role in neurological disorders including AD, thus the analysis of cellular and molecular composition in CSF and peripheral blood in comparison with the CNS system provides invaluable information to biological and/or disease processes of AD.

At the late stage of AD (post-mortem), we detected increased expressions of markers involved in phagocytosis in the CP macrophages, whereas myeloid cells in the GFM increased the don´t eat me signal CD47 and its receptor CD172a (SIRPα). At this stage, microglia served most likely as a source of MIP-1β (CCL4), a ligand for CCR5 and chemoattractant inducing migration of phagocytic macrophages into the brain. Our findings are in line with the concept of dystrophic microglia in late-onset AD, in which microglia are unable to remove aggregated amyloid, present as an exhausted phenotype and with exacerbated aging-dependent microglia deterioration[50].

In conclusion, based upon our findings, we propose that myeloid cells in the CSF present activation phenotypes (e.g. changes in bioenergetic pathways, phenotypic changes and increase phagocytic activity) which may help our system to defend against pathologic stimuli and/or to regulate the activation of adaptive immunity. Once this process becomes dysregulated (such as in AD), it could lead to chronic inflammation in different compartments, which could further harm the system, such as dysregulation of T cell activation and colonization[51]. At

the late stage of the disease (post-mortem), we detected increased inhibitory signalling on myeloid cells and exhausted microglia with inflammatory phenotypes. However, numerous open questions remain unanswered, including (1) what is the biological function of P2Y$_{12}$$^+$ myeloid cells in the CSF and CP, and whether these cells are recruited from the brain or the peripheral blood? (2) Which soluble factors in CSF and/or CP drive the differentiation of the myeloid cell population? On one hand, these open questions can be hardly answered by using animal models, due to large differences in immune cell regulation and differentiation between species, on the other hand, solving these questions using human systems is ethically and technically challenging. To our opinion, development of a proper ex vivo/in vitro model of human system (such as organoid culture) consisting of circulation would serve as a promising system for studying this complex interaction between the peripheral and the central system.

## Methods

This study complies with all relevant ethical regulations and was approved by the Ethics Commission of Charité–Universitätsmedizin Berlin (Ethikkommission der Charité–Universitätsmedizin Berlin; registration number EA1/187/17 and EA1/241/17), Berlin, Germany. All study participants provided informed consent before any study-related procedures were undertaken. All participants received no compensation.

### Human blood and CSF samples

Venous blood and lumbar cerebrospinal fluid (CSF) samples ($n = 117$) were obtained from control individuals ($n = 21$) or patients with AD ($n = 18$), MCI ($n = 7$), HD ($n = 46$), depression ($n = 11$), FTLD ($n = 5$) and SCZ ($n = 9$) (Supplementary Table 1). PBMCs were isolated from EDTA-blood (20 ml) within 1 h of the blood draw through Biocoll (Biochrom GmbH, Berlin, Germany) density centrifugation at $1200 \times g$ for 20 min at room temperature. The blood mononuclear cell fraction was recovered and washed twice in phosphate-buffered saline (PBS; Biochrom GmbH) at $300 \times g$ for 10 min. For the isolation of CSF cells, CSF was centrifuged once at $300 \times g$ for 10 min (4 °C). The cell pellet (PBMCs or CSF cells) was then fixed with fixation/stabilization buffer (SmartTube) and frozen at −80 °C until analysis by mass cytometry. PBMCs and CSF cells of a total of eleven CON, eight AD, seven MCI and twelve HD out of those 117 samples were used for CyTOF analysis.

### Human brain autopsy

Human brain tissue was obtained through the Netherlands Brain Bank (www.brainbank.nl). The Netherlands Brain Bank received permission to perform autopsies and to use tissue and medical records from the Ethical Committee of the VU University medical center (VUmc, Amsterdam, The Netherlands). All donors have given informed consent for autopsy and use of their brain tissue for research purposes. Generally, the autopsies of frontal lobe (Gyrus frontalis medius, GFM) and choroid plexus were performed within 4–10 h after death (Supplementary Table 1). Brain tissue collected

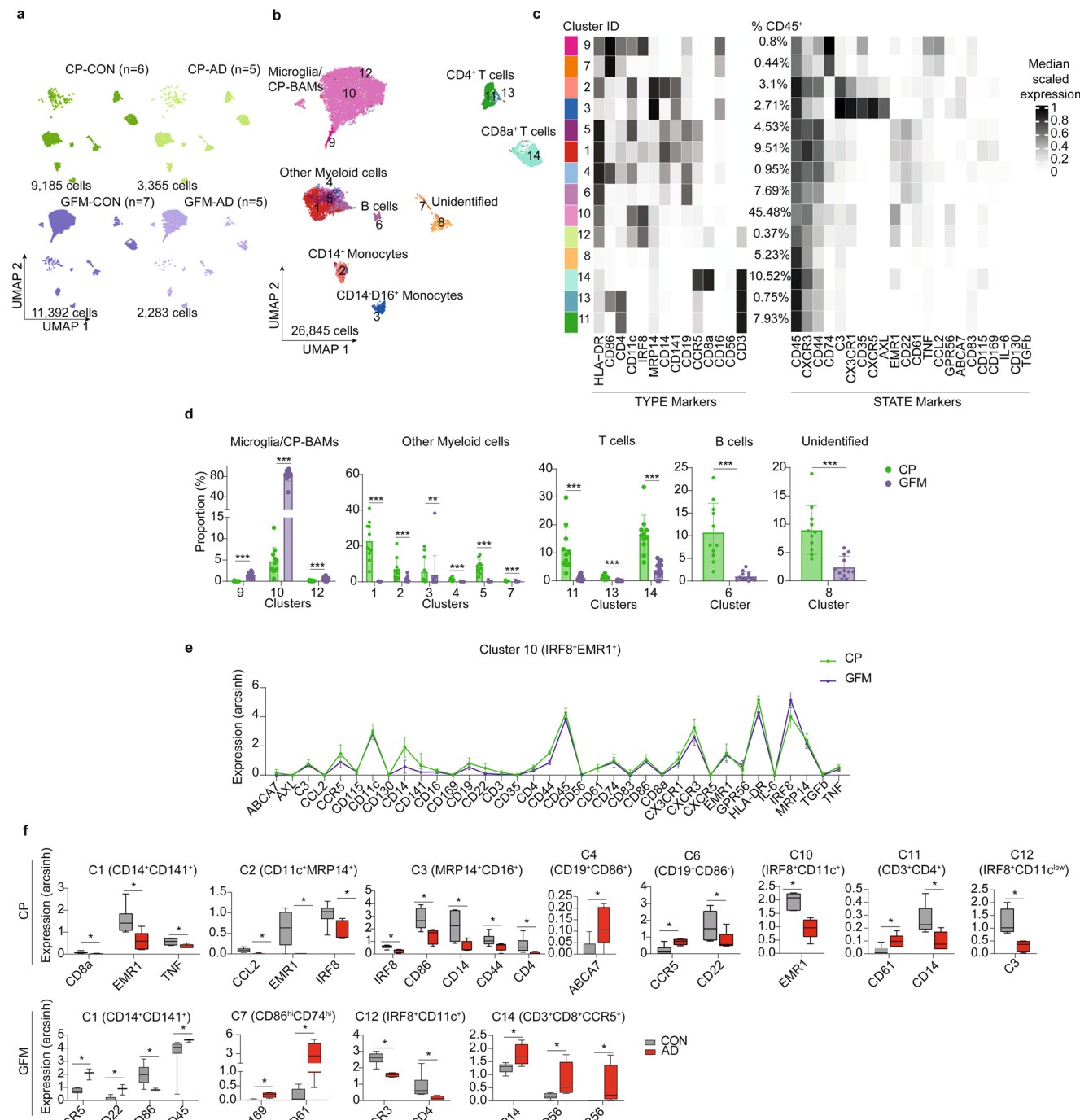

**Fig. 7 | Differential heterogeneity of CD45⁺ immune cells in CP and GFM of AD donors, compared to the CON donors, determined by using *Panel A* (CP: CON, *n* = 7; AD, *n* = 4, GFM: CON, *n* = 7; AD, *n* = 5, biological replication). a** Overlaid UMAP plots of CP and GFM samples from AD (green dots) and CON (purple dots) donors. **b** UMAP plots of all samples. The colouring indicates 14 clusters representing diverse immune cell phenotypes, defined by the *FlowSOM* algorithm. **c** Phenotypic heatmap of cluster identities depicting the expression levels of 14 TYPE markers used for the cluster analysis and 22 STATE markers. Heat colours of expression levels have been scaled for each marker individually (to the 1st and 5th quintiles) (black, high expression; white, no expression). **d** The bar graphs show differentially abundant clusters between CP and GFM. Data displayed as mean ± SD.

An FDR-adjusted *p*-value < 0.05 was considered statistically significant, determined using the edgeR test for differential cluster abundance (*$p < 0.05$; **$p < 0.01$; ***$p < 0.001$). **e** The graphs showed differential marker expression (mean ± sd) between the CP^epi^-BAM in CP and microglia in GFM (both are defined as cluster 10). FDR-adjusted Mann–Whitney *U*-test, two-sided; *$p < 0.05$ is considered significant. **f** The Box plots shows differential marker expression of immune cells in CP or in GFM of AD donors, compared with the CON donors, using Mann–Whitney *U*-Test, two-sided. The *p*-value < 0.05 (*) is considered statistically significant. Boxes extend from the 25th to 75th percentiles. Whisker plots show the min (smallest) and max (largest) values. The line in the box denotes the median. Each dot represents the value of each sample. Source data (for **d**, **e** and **f**) are provided as a Source Data file.

for this study was from the donors whose post-mortem CSF was between pH 5.9 and 6.9 (Supplementary Table 1). An overview of the donor information and post-mortem variables is summarized in Supplementary Table 1.

## Human brain immune cell isolation
The isolation was started within 2–25 h after autopsy. Approximately, 2–10 grams tissue was first mechanically dissociated through a metal sieve in a glucose-potassium-sodium buffer (GKN-BSA; 8 g/l NaCl,

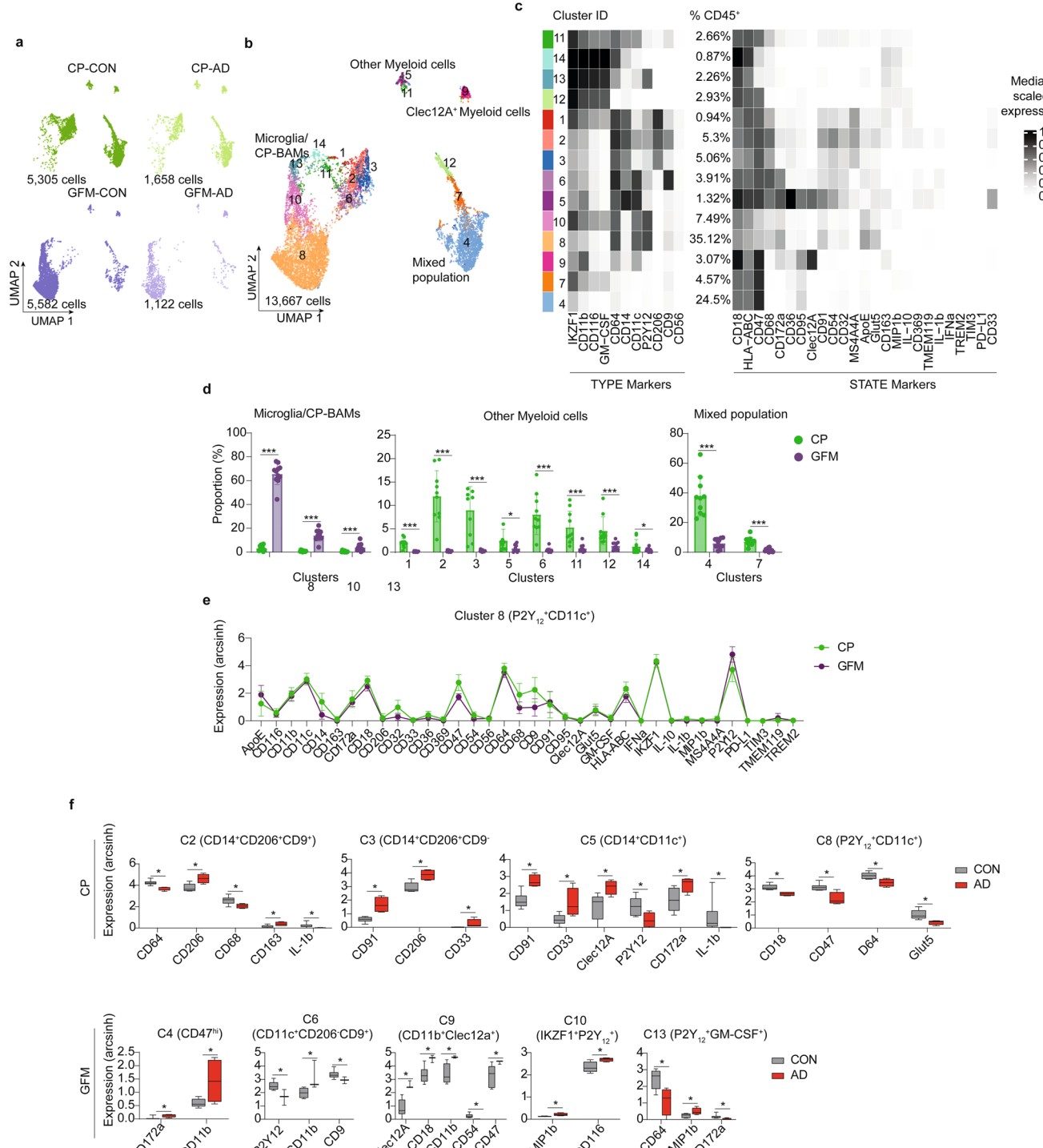

**Fig. 8 | Differential heterogeneity of CD45+ immune cells in CP and GFM of AD donors, compared to the CON donors, determined by using *Panel B (CP:* CON, *n* = 7; AD, *n* = 3, *GFM*: CON, *n* = 7; AD, *n* = 4, biological replication). a** Overlaid UMAP plots of CP and GFM samples from AD (green dots) and CON (purple dots) donors. **b** UMAP plots of all samples. The colouring indicates 14 clusters representing diverse immune cell phenotypes, defined by the FlowSOM algorithm. **c** Phenotypic heatmap of cluster identities depicting the expression levels of 10 TYPE markers used for the cluster analysis and 25 STATE markers. Heat colours of expression levels have been scaled for each marker individually (to the 1st and 5th quintiles) (black, high expression; white, no expression). **d** The bar graphs show differentially abundant clusters between CP and GFM. Data displayed as mean ± SD.

An FDR-adjusted *p*-value < 0.05 was considered statistically significant, determined using the edgeR test for differential cluster abundance (*$p$ < 0.05; **$p$ < 0.01; ***$p$ < 0.001). **e** Line graphs showing differential marker expression (mean ± sd) between the CP$^{epi}$-BAM in CP and microglia in GFM (both are identified as cluster 8). FDR-adjusted Mann−Whitney *U*-test, two-sided; *$p$ < 0.05 is considered significant. **f** The Box plots shows differential marker expression of immune cells in CP or in GFM of AD donors, compared with the CON donors, using Mann−Whitney *U*-Test, two-sided. The *p*-value < 0.05 (*) is considered statistically significant. Boxes extend from the 25th to 75th percentiles. Whisker plots show the min (smallest) and max (largest) values. The line in the box denotes the median. Each dot represents the value of each sample. Source data (for **d**, **e** and **f**) are provided as a Source Data file.

0.4 g/l KCl, 1.77 g/l $Na_2HPO_4.2H_2O$, 0.69 g/l $NaH_2PO_4.H_2O$, 2 g/l D-(1)-glucose, 0.3% bovine serum albumin (BSA, Sigma-Aldrich); pH 7.4). The samples were then supplemented with collagenase Type I (3700 units/ml; Worthington, Lakewood, NJ, USA) and DNase I (200 µg/ml; Roche Diagnostics GmbH) for 1 h at 37 °C while shaking. Cell suspension was put over a 100 µM cell strainer and washed with GKN-BSA buffer before the pallet was resuspended in 20 ml GKN-BSA buffer. Next, 10 ml of Percoll (Amersham, GE Healthcare) was added dropwise and tissue was centrifuged at $3220 \times g$ for 30 min (4 °C). Three different layers appeared: upper layer containing myelin, a lower erythrocyte layer and the middle layer containing all cell types including microglia. The middle layer was carefully taken out without disturbing the myelin layer and washed first with GKN-BSA buffer, followed by cell sorting using flow cytometry.

## Fluorescence-activated cell sorting

The single-cell suspension was incubated with an FC receptor blocking reagent (Miltenyi Biotec, 5170126102) (1:20) and anti-CD45 (eBioscience, 11–9459) antibodies (1:20) at 4 °C for 15 min. Cells were washed twice and suspended in glucose-potassium-sodium buffer (GKN-BSA; 8 g/L NaCl, 0.4 g/L KCl, 1.77 g/L $Na_2HPO_4.2H_2O$, 0.69 g/L $NaH_2PO_4.H_2O$, 6 g/L D-(1)-glucose, pH 7.4) with 0.3% BSA. 7-AAD (BD Biosciences, 5168981E) (1 µg/mL) was added for cell death detection. Cells were filtered over a 70 µm cell strainer before sorting. Cells were sorted/gated that were alive, single, and CD45$^+$ with the FACSAriaIII. The sorted CD45$^+$ fraction (Supplementary Fig. 5) was placed on ice, centrifuged at 300xg for 5 min, supernatant removed, and pellet was fixed with fixation/stabilization buffer (SmartTube) and stored at −80 °C.

## Barcoding

**Live cell barcoding.** Individual CSF samples ($0.5–1 \times 10^4$ cells) were pelleted and stained with $^{89}$Y-CD45 (Fluidigm) for 30 min at 4 °C. Cells were then washed and pooled with PBMCs from the same individual.

**Intracellular barcoding.** After fixation and cryopreservation (at −80 °C), cells were thawed and subsequently stained with premade combinations of six different palladium isotopes: $^{102}$Pd, $^{104}$Pd, $^{105}$Pd, $^{106}$Pd, $^{108}$Pd & $^{110}$Pd (Cell-ID 20-plex Pd Barcoding Kit, Fluidigm). This multiplexing kit applies a 6-choose-3 barcoding scheme that results in 20 different combinations of three Pd isotopes. After 30 min staining (at RT), individual samples were washed twice with cell staining buffer (0.5% bovine serum albumin in PBS, containing 2 mM EDTA). Total of up to 20 samples were pooled together, washed and further stained with antibodies.

**Antibodies.** Anti-human antibodies (Supplementary Table 2–7) were purchased either pre-conjugated to metal isotopes (Fluidigm) or from commercial suppliers in purified form and conjugated in house using the MaxPar X8 kit (Fluidigm) according to the manufacturer's protocol.

## Surface and intracellular staining

After cell barcoding, washing and pelleting, the combined samples were resuspended in 100 µl of antibody cocktail against surface markers (Supplementary Table 2–7) and incubated for 30 min at 4 °C. Then, cells were washed twice with cell staining buffer. For intracellular staining, the stained (non-stimulated) cells were then incubated in fixation/permeabilization buffer (Fix/Perm Buffer, eBioscience) for 60 min at 4 °C. Cells were then wash twice with permeabilization buffer (eBioscience). The samples were then stained with antibody cocktails against intracellular molecules (Supplementary Table 2–7) in permeabilization buffer for 1 h at 4 °C. Cells were subsequently washed once with permeabilization buffer, then were washed again with PBS before overnight incubation in 2% methanol-free formaldehyde

solution (FA). Fixed cells were then washed and resuspended in 1 ml of 500 nM iridium intercalator solution (Fluidigm) for 1 h at RT. Next, the samples were washed twice with cell staining buffer and then twice with $ddH_2O$ (Fluidigm). Cells were pelleted and kept at 4 °C until CyTOF measurement.

## Mass cytometry data processing and analysis

As described previously[22,26,27], Cytobank (www.cytobank.org) was used for initial manual gating on single cells and boolean gating for de-barcoding. Nucleated single intact cells were manually gated according to DNA intercalators $^{191}$Ir/$^{193}$Ir signals and event length. For de-barcoding, Boolean gating was used to deconvolute individual sample according to the barcode combination. All de-barcoded samples were then exported as individual FCS files for further analysis. Each FCS file was compensated for signal spillover using the R package CATALYST[52] and transformed with arcsinh transformation (scale factor 5) prior to data analysis. For further analysis we used a previously described scripts and workflows[25]. Only samples with >50 cells were considered for the downstream data analysis. For a first assessment of the data we obtained multi-dimensional scaling (MDS) plots based on median marker expression from all markers in order to assess overall similarities between samples and/or groups/conditions. We next obtained a marker ranking identified by the PCA-based non-redundancy score (NRS) which can give us an idea of which markers explain most of the variability among samples. For following analysis, we selected the ten highest scoring NRS markers plus other lineage markers not included among this, and defined them as "TYPE" makers, thus the ones which will be used for unsupervised clustering. The rest of the markers were defined as "STATE" markers, which will be used to better describe the different populations and further assess differences in cell activation status between groups. Unsupervised clustering was performed using the FlowSOM[28]/ConsensusClusterPlus[29] algorithms which are included in the CATALYST package. We then selected the number of meta-clusters used for further analysis based on the delta area plots (which asses the "natural" number of clusters that best fits the complexity of the data) together with visual inspection on the phenotypic heatmaps with an aim to select a cluster number with consistent phenotypes that would also allow us to explore small populations. For dimensionality-reduction visualization we generated UMAP representations using all markers as input and down-sampled to a maximum of 1000 cells per sample.

## Cell culture and stimulation

PBMCs were resuspended in 1 mL of RPMI1640 (Biochrom GmbH, Berlin, Germany) containing 10% heat-inactivated foetal calf serum (FCS) (Sigma-Aldrich, St. Louis, USA), penicillin (100 U/mL; Biochrom GmbH, Berlin, Germany) and streptomycin (100 µg/ mL; Biochrom GmbH, Berlin, Germany). Cell concentration was adjusted to ~ $2 \times 10^6$ cells/ mL. About $2 \times 10^5$ cells (per well) were transferred into ultra-low-attachment 96-well plate (Corning, New York, USA). Cells were treated with either PBS, CSF (20% v/v), IL-8 (50 ng/ml), MIP-1α (10 ng/ml), LPS (100 ng/mL) or both CSF and LPS. To inhibit protein transport from Golgi apparatus to the endoplasmic reticulum, monensin (5 µg/mL; BioLegend, San Diego, USA) was also added. After 4–6 h incubation, cells were harvested, fixed and stored at −80 °C until CyTOF analysis.

## Flow cytometry analysis

Cryopreserved and fixed PBMCs (4x CON, 4x MCI and 4x AD) were thawed and washed twice in staining buffer (PBS containing 0.5% BSA and 2 mM EDTA). The cells were incubated in Fc Block for 10 min at 4 °C. Cells were then stained for CD14 (PerCP-Cy5.5, clone HCD14), HLA-DR (APC-Cy7, clone L243), CX3CR1 (FITC, clone 2A9-1) and P2Y$_{12}$-Biotin for 20 min at 4 °C in staining buffer. Cells were washed once in staining buffer and then incubated in PE-Cy7-Streptavidin for 20 min at 4 °C. For the intracellular staining, the samples were washed once with

staining buffer, then fixed in 2% FA for 30 min at 4 °C. After incubation, the samples were washed with permeabilization buffer (eBioscience), and subsequently stained with IL-8 (PE, clone E8N1), MIP-1α (APC, clone CCL3) and TNF (BV421, clone Mab11) for 30 min at 4 °C in permeabilization buffer. Stained cells were subsequently washed once with staining buffer. Cellular fluorescence was assessed with CantoII (BD FACSDiva Software 6.1.3; BD Biosciences) and data were analyzed with FlowJo software (10.4.2) (TreeStar) and GraphPad Prism 9. Forward- and side-scatter parameters were used for exclusion of doublets from analysis (Supplementary Fig. 4).

## Luminex

Cytokine levels in plasma and CSF were measured using cytokine protein multiplex assay (MILLIPLEX® Multiplex Assays, Merck KGaA) on a Luminex 200 platform and Bio-Plex Manager 6.2 software (Bio-Rad Laboratories GmbH) according to the manufacturer's instructions. Plasma and CSF samples ($n = 117$) of control individuals ($n = 21$) or patients with AD ($n = 18$), MCI ($n = 7$), HD ($n = 46$), depression ($n = 11$), FTLD ($n = 5$) and SCZ ($n = 9$) were used for Luminex analysis (Supplementary Table 1).

## Monocyte isolation and 1,2-$^{13}$C$_2$-glucose experiment

Frozen human peripheral blood mononuclear cells (PBMCs) were thawed, washed and pooled in MACS buffer (0.5% BSA in PBS containing 2 mM EDTA). Monocytes were isolated using negative selection, pan-monocyte Isolation Kit (Miltenyi Biotec, Bergisch Gladbach, Germany) according to manufacturer's specifications. Briefly, PBMCs were resuspended in MACS buffer. FcR blocking reagent and biotin-antibody cocktail were added, mixed thoroughly and incubated at 4 °C. After 5 min incubation, MACS buffer and anti-biotin micro beads were added and incubated at 4 °C for 10 min. Stained cells were then washed with MACS buffer and pelleted (4 °C, 300 × g, 8 min). The pellet was then resuspended in MACS buffer and loaded onto the MACS column. The column was then washed twice with MACS buffer. The flow-through and washed fraction containing unlabelled monocytes was collected. Cell number and viability were determined by 0.2% trypan blue staining. Monocytes were cultured and stimulated in the presence of 1,2-$^{13}$C$_2$-glucose using the protocol described above. Cells were then harvested, pelleted and shock-frozen in liquid nitrogen. A 100 μl of a mixture of acetonitrile and water (H$_2$O:ACN (1:1)) was then added to the frozen pellet and incubated on ice for 5 min. Cell lysate was then centrifuged at 15,000 × g for 10 min (4 °C). A 75 μl of supernatant was stored at −80 °C until HPLC-MS/MS analysis.

## HPLC-MS/MS

Cell lysate in H$_2$O:ACN (1:1, as described above) was thawed and measured by HPLC-MS/MS. To reduce the chelating interactions between citrate or phosphates, and a broadening of the chromatographic peaks, which is caused by the metallic part of the instrument, the passivation of the system was performed prior to the analysis of targeted metabolites with HPLC-MS/MS. To do so, the whole system was washed with 0.5% H$_3$PO$_4$ in ACN:H$_2$O (9:1) overnight, followed by 1 hr of H$_2$O and 2 hr of column conditioning with the chosen mobile phases (*mobile phase A*: 10 mM CH$_3$COONH$_4$, pH 9 in H$_2$O with 5 μM Agilent InfinityLab deactivator additive; *mobile phase B*: 10 mM CH$_3$COONH$_4$ (aq.) pH 9 in ACN with 5 μM Agilent InfinityLab deactivator additive). Both mobile phases were filtered before use with a 0.2 μm filter.

The analysis was conducted with an Agilent 1290 Infinity II HPLC system, coupled with an Agilent 6495 QqQ mass spectrometer, with an Agilent jet stream source with electrospray ionization (AJS-ESI). *Method*: dynamic multiple reaction monitoring (dMRM). *Ionization mode*: positive and negative. *Sheath gas flow*: 12 L/min for both, positive and negative modes. *Sheath gas temperature*: 350 °C for both, positive and negative modes. *Capillary*: 4500 V for positive mode and 3500 V for negative. *Nozzle voltage*: 750 V for positive mode and 0 V for negative. *Drying gas temperature*: 210 °C for both, positive and negative modes. *Drying gas flow*: 20 L/min for both, positive and negative modes. Nebulizer 30 psi for both, positive and negative modes. *Funnel*: High P RF 190 and Low P RF 40 for positive mode and High P RF 110 and Low P RF 60 for negative. *Acquisition software*: MassHunter 10 Acquisition software G3335 (Agilent Technology, Waldbronn, Germany).

The chromatographic separation was performed with an Agilent Poroshell 120 HILIC-Z peek-lined column, at 30 °C, and with a gradient starting from 90% B to 60% B in 10 min and a total run time of 21 min. The autosampler was kept at 4 °C to preserve the samples and the injection volume was 1 μL. The MS parameters and the targeted transitions were optimized and analysed with the Agilent MassHunter Optimizer software (MassHunter 10 Acquisition software G3335, Agilent Technology), and the chosen acquisition mode was dMRM to increase the sensitivity. An external calibration curve of standards in H$_2$O:ACN (1:1) with at least 8 levels was injected before the analysis of the samples as well as QCs every 10–15 injections to evaluate the stability of the system during the analysis. The method was validated based on the ICH guideline M10 on bioanalytical method validation. The quantitation was performed by external calibration of standards in H$_2$O:ACN (1:1), as mentioned above. We evaluated the specificity and the selectivity, the matrix effect and the recovery in peripheral blood mononuclear cells (PBMCs), the accuracy and the precision, the carry-over, and the short- and long-term stability. The detailed procedure of $^{13}$C$_2$-glucose tracing experiment and HPLC-MS/MS quantification was recently published and freely available[53].

## Statistics and reproducibility

No randomization and blinding strategies were applied in this study. However, data processing and analysis, as well as statistical testing were carried out in an unsupervised manner. Quantitative data are shown as independent data points with median or Box-Whisker-Plot. Differential analysis of cell population abundance between groups were performed using EdgeR[54] available through the R package diffcyt[55] (with default parameters, and filtering parameters set to minimum number of cells = 3 in at least minimum number of samples = number of samples in each group) and false discovery rate (FDR) adjustment (at 5% using Benjamini-Hochberg procedure) for multiple hypothesis testing. Unless otherwise stated, significant differences in marker expression between clusters or between groups were calculated using Mann–Whitney $U$-test (5% FDR) in GraphPad Prism (version 9).

## Reporting summary

Further information on research design is available in the Nature Portfolio Reporting Summary linked to this article.

## Data availability

The CyTOF data generated in this study have been deposited in the FlowRepository database under accession code FR-FCM-Z5XD. The HPLC-MS/MS data generated in this study have been deposited in the refubium database (https://refubium.fu-berlin.de) under accession link https://doi.org/10.17169/refubium-36351. Source data are provided with this paper as Source Data file. Source data are provided with this paper.

## Code availability

Codes used for CyTOF data analysis in this study are previously published by Crowell H et al. 2022 and available on https://github.com [https://github.com/HelenaLC/CATALYST].

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

## Acknowledgements

We thank Jasmin Jamal El-Din and Christian Böttcher for excellent technical assistance with sample collection and ex vivo experiments. We would also like to acknowledge the assistance of the Flow & Mass Cytometry Core Facility (BIH at Charité—Universitätsmedizin Berlin, Germany) and the Netherlands Brain Bank (Amsterdam, The Netherlands). G.Gi. was funded by the Elsa-Neumann scholarship of the State of Berlin. G.Gi. and G.Ga. received the PhD scholarship from the NeuroMac School (DFG, the German Research Foundation—Project-ID 259373024—CRC/TRR 167 (B05)). A.M. and C.D. were were funded by the Deutsche Forschungsgemeinschaft (DFG, the German Research Foundation—Project-ID 259373024—CRC/TRR 167 (B12)), and the Leducq Foundation (19CVD01). R.E.v.D., J.M. and E.M.H. were funded by ZonMW 733050107. C.B. and J.P. were funded by the Deutsche Forschungsgemeinschaft (DFG, the German Research Foundation—Project-ID 259373024—CRC/TRR 167 (B05 and B07)).

## Author contributions

C.B. and J.P. conceived and designed the project. C.B., C.F.Z. and D.K. designed the antibody panels for mass cytometry. G.Gi. and M.K.P. performed the metabolomic tracing experiments using [13]C-labelled glucose. E.J.S. and J.P. recruited the patients and provided peripheral blood and cerebrospinal fluid samples, as well as the patients´ clinical data. J.M., R.E.v.D. and E.M.H. performed single-cell isolation, sorting and cryopreservation of the immune cells from post-mortem brain tissues. C.B., C.F.Z., G.Ga., J.K.H.L. and S.S. analyzed and interpreted the data. A.D. and G.Gi. performed the ex vivo experiment. A.D., C.D. and A.M. were responsible for Luminex analysis. C.B., C.F.Z., S.S., F.P. and J.P. wrote the manuscript.

## Funding

## Competing interests

The authors declare no competing interests.
