## [Peer Review File · Nature Communications]

Differential compartmentalization of myeloid cell phenotypes and responses towards the CNS in Alzheimer's diseaseREVIEWER COMMENTS

Reviewer #1 (Remarks to the Author):

This manuscript from Zapta et al. titled, "Differential compartmentalization of myeloid cell phenotypes and responses towards the CNS in Alzheimer's disease," explores immune cell phenotypic differences between the blood and cerebrospinal fluid (CSF) in healthy adult patients and those with Alzheimer's disease (AD). The authors provide data showing phenotypic differences in immune cell populations between the blood and CSF in both control and AD patients. The author's applied numerous state-of-the-art techniques to probe immune cells in these compartments and have generated an impressive dataset. However, although there may be potentially interesting results, the manuscript is overwhelming and does not highlight key findings. The figures are overcrowded and the results at times are overinterpreted since data are mostly descriptive and functional characteristics and consequences remain unexplored. Below are specific key points for each figure that should be addressed.

Figure 1:

- Several data plots are shown for each analysis, but most are glossed over without discussion or explanation. The authors should discuss each plot shown if it is to be included. For example, Fig.1B shows NRS values for several genes, but these data are better expressed in Fig.1D,E, which shows the expression of these variable genes in different cell clusters.
- The terms "TYPE" and "STATE" markers are used throughout the manuscript but are never defined.
- Fig.1C is not referenced properly within the text.
- Measures of "proportion" are misleading. For example, the authors claim, "The CSF-enriched clusters were mainly identified as myeloid cells (Clusters 13 and 15);" however, Cluster 13 was only expressed in two samples. Moreover, these clusters are not easily visible on the UMAP plots. Are these "proportions" being driven simply by a significantly lower number of total cells in the sample? Raw cell yields for each sample and resulting clusters should be reported.
- "In addition, a differentially abundant cluster CD4+ T cell (Cluster 17) was detected at a higher frequency in the CSF." The term "higher frequency" is misleading since the raw numbers are significantly lower in the CSF.
- Actual phenotyping results are presented in passing. "Overall, compared to classical monocytes (Cluster 16), CSF-enriched myeloid cells showed higher expression of markers involved in inflammatory responses, phagocytosis and metabolism... whereas MRP14, CD14, CD35, EMR1, CD38, CD369 (Clec7A) and TNF were expressed at a lower level." These seem like the results that should be discussed more in-depth here. Additionally, TNF is reported to be expressed at a lower level, but "inflammatory responses" are reported to be higher. This warrants discussion.
- In panel 1G, it seems like there are more samples for PBMC-CON than CSF-CON. They should be equal (N=11) since taken from the same individual. Were samples excluded?

Figure 2:

- The authors aim "to prove the existence of this rare cell population," referring to the "rare population of myeloid cells in the CSF with a transcriptomic signature matching microglia," which are reported to be "found only in the CSF of subjects with neuroinflammation," by performing "another CyTOF measurement of the same CON donors." Why are controls being used if this is an inflammatory population? Also, is this population not CD16+/CD14 low non-classical monocytes which have been shown to be enriched in CSF compared to blood?
- Though the "CSF-enriched" cells discussed can be identified on the UMAP plots, they do not seem to cluster separately based on expression patterns.
- Additional microglia signature markers should be assessed (e.g., TMEM119, SALL1, TGFBR1) before determining these cells are "matching microglia".
-

Figure 3:

- Are the clusters in Fig.3 the same as in the previous figures? If so, why are they being compared separately from CON? These should have CON in the same figure as reference to evaluate changes with disease condition. If not, how are these clusters being generated, and how can they be compared to the previous clusters?
- The authors conclude there are "different abundances in myeloid, NK and lymphoid cell clusters in CSF compared to the peripheral blood"; however, the setup of Fig.3B divides the analysis into disease state (i.e., MCI v. AD v. HD). These distinctions are not discussed in the text.

Figure 4:

- IL-10 is introduced in text, but all other references claim it was IP-10.
- Confusing text: "The Luminex assay revealed a higher concentration of IL-8, MIP- β , CCL2 (MCP-1), IL-6 and IP-10 in the CSF, whereas the level of the plasma TNF and Rantes (CCL5) were higher than in the CSF."
- Labels for Fig.4C need work. Y-axis should read, "CSF MIP- α Concentration (pg/mL)" or "CSF IL-6 Concentration (pg/mL)"; X-axis should read, "CSF IL-8 Concentration (pg/mL)"; and Legend should read, "CON", "MCI", "AD", and "HD".
- Why are data for FTD and SCZ not presented?

Figure 5:

- The biological relevance of ex vivo LPS treatment is not clear. First, how is CSF being isolated/applied? Is the CSF depleted of cells? Moreover, LPS will seldom-if-ever directly stimulate cells in CSF. Another more biologically relevant assay should be employed here.
- Which CSF is used in "CSF-only" and "CSF+LPS" treatment? The text claims both CON-CSF and AD-CSF are used. Are these mixed?
- The confusion introduced by the two points above make interpretation of the rest of the figure difficult if not impossible.
- More work would be needed (not just relying on CCR2 and P2RY12) to conclude Cluster 15 is similar to the population in fig 2.

Figure 6:

- Authors should use consistent terms between text and figures (e.g., pyruvate v. pyruvic acid; lactate v. lactic acid).
- For Fig.6E, does the decreased "ratio-pyruvate-to-lactate" not indicate increased lactate production (i.e., increased value of the denominator)? The text seems to interpret these data backwards.

Figure 7/8:

- Fig.8E is thrown into the middle of the discussion of Figure 7. Moreover, this is the only panel discussed for Figure 8. Authors should discuss all data presented in figures.
- Discussion of these data ends on a very weak note: "Together, it is tempting to speculate..."

General:

- The authors repeatedly write, "We asked whether... To prove this assumption..." but no assumption is being clearly made. Changing this to, "To explore this..." or another phrase may make the thought process of the experimenters easier to follow.
- The figures are overcrowded

Reviewer #2 (Remarks to the Author):

The authors Zapata et al., have presented an interesting study on the myeloid cell phenotypes in healthy and AD individuals. However, the study needs some additional analysis and clarification that are suggested here:

1. The authors have provided donor and patient information. But details like the number

of males and females in this study, APOE status of the individuals, Braak or CERAD, or cognitive assessment scores are not mentioned. The authors should include this information in the text as well as the supplementary.

2. Age as well as sex influence the neuroinflammatory response. The authors should include analysis by stratifying the samples based on the age and sex of the individuals considered in this study.

3. Can the authors describe more about the TYPE and STATE markers? How are they different? The median scaled expression of STATE markers in both Figures 1 and 2 is lesser than TYPE markers. So, is there a significance of STATE markers if we consider the expression values?

4. How much was the difference in the cell numbers in CSF and PBMC? Based on Figure 1C, PBMC has more clusters and cells than CSF. Was cell count normalization carried out before comparing the markers between the groups?

5. Lipid species have been studied in detail as they are associated with neuroinflammatory responses. Did the authors identify lipid markers in their study?

6. In figures 3 and 4, the differential marker expression is similar in AD and MCI samples. Does it suggest that the neuroinflammatory response is similar in both disease phenotypes?

7. Can the results presented here use for building predictive models for the onset of AD?

8. In Supplementary Table 1, samples for depression, FTL, and SCZ are indicated. Have these been used for comparative analysis?

9. "Venous blood and lumbar cerebrospinal fluid (CSF) samples were obtained from control individuals or patients with neurological disorders (Supplementary Table 1)." Were the samples collected only once or more than one time from these individuals?

Minor comment:

In the abstract, kindly rephrase "health" to "healthy"

Reviewer #3 (Remarks to the Author):

Fernández Zapata and colleagues provide a detailed analysis of multiple human compartments and cell types with various high-dimensional technologies. In their study, they comprehensively characterized human-derived samples of blood, cerebrospinal fluid, choroid plexus, and brain parenchyma while focusing on immune cell abundance and phenotype. The authors used several algorithms to analyze the generated data and interpret it. It is important to appreciate the extent of work done in this study due to the use of human samples and their limited availability, especially those derived from the central nervous system. As the authors mentioned, using human systems is ethically and technically challenging.

While the manuscript is intriguing, the majority of the data is descriptive, and some technical and conceptual factors are missing, along with some issues in data interpretation. Additionally, data describing the composition of human blood, cerebrospinal fluid (CSF), choroid plexus, and brain parenchyma are available in current literature (with same or other analysis methods; PMID: 33239300, PMID: 33239300), thus limiting the novelty of this study. Moreover, the main conclusions are mostly hypothetical, as also stated by the authors. Collectively, the authors do provide a comprehensive depiction of cells and secreted factors, at the protein level, between different compartments, yet they mostly rely on one method (mass cytometry) and no validations were used for key results.

Major comments:

1. Comparing different cell clusters, meaning different cell subsets, will most likely provide significant differences between the clusters. Results describing cluster-specific phenotypes, although interesting and valuable to understand the nature of each cell subset, should not take so much focus. It is the group-specific differences that need to

be thoroughly addressed. It is also somewhat confusing to show significant differences between clusters along with other group-based analyses.

2. Tissue-related comparisons are another similar example of results that are important to understand the cell environment in each compartment, yet should be used for supporting the main data rather than being it. For example, besides further validating that CSF and plasma have different compositions, what does the data in Figure 4a provide us? It would be more interesting to see the levels of each inflammatory mediator for each group (control, AD, MCI, HD, FTL, depression and schizophrenia), including those that were not included in Figure 4b.

3. An important variable in this study that is not referred to by the authors is donor age, some of the differences may be age-related, as the mean age for the CON group is 62 yr while in the AD group it is 72 yr. That 10 year gap may be a significant contributing factor of inflammation. Furthermore, if possible, please also refer to BMI and other relevant parameters of donors. On the same note, are there any gender-specific differences that could be identified in the various compartments?

4. It was confusing and laborious to go back and forth with several of the figures due to the split layout according to panel. Might be better to have the data side by side according to plot type and indicate the relevant panel below each part or in the figure legend.

5. While the manuscript is mostly well-written (with minor typos), the results section contains too many technical notes (e.g., page 5). It would benefit the reader to pare out as much of this as possible. This information can be included in figure legends and methods, and thus the paper will be much more readable. Negative data could be deemphasized by significantly shortening these sections.

Minor comments:

1. The manuscript is lacking gating examples for CyTOF gating (e.g. live single-cell gating). Also, manual gating to some of the results would provide some reassurance to the results generated by some of the algorithms/packages; some results rely on a very limited number of cells and their inference might then be revised (for example, cluster 18 in Figure 4d). Accordingly, referring to cell counts (per subset/analysis) would support the validity of the data. Some CSF subsets/clusters likely have less than 10 cells; is it possible that some of the CSF-PBMC cluster proportion differences were created by the large difference in total cell counts between these compartments? (perhaps downsample). Please also provide a gating example for cell sorting according to tissue.

2. Many plots indicate "Expression" on the Y-axis with no units, please clarify. Can the authors elaborate on what is a reasonable expression level? For example, does the small difference in expression level (~ 0.1) in Figure 7f Cluster 4 CP has a biological meaning? Please explain for other plots with low expression values.

3. Please indicate "Proportion (%)" out of what population/pool of cells (i.e., all CD45+ live single cells?).

4. Some of the presented UMAPs lack the total number of cells and the number per condition/tissue. This is especially important in CSF samples that in some cases include very few cells.

5. It would benefit the reader to have cluster annotations in each heat map, along with cluster numbers over each dimensionality reduction map (color blind compatible).

6. Please clarify the exact details of "cell culture and stimulation". For example, how was the volume of CSF determined in the in vitro experiments?

- 7. For Figure 5, the authors mention that "PBMCs showed changes in phenotype when treated with CSF", however, the MDS plot does not show much difference between no stimulation and CSF. Small if any differences are also evident in Figure 5c. This lack of difference is also surprising compared to the results in Figure 6, showing a significant change when CSF is added.**
- 8. In Figure 8, since all pooled samples were not analyzed in the same CyTOF run, how was signal intensity normalized between samples/runs?**
- 9. In the in-house generated CyTOF antibodies, how was optimal antibody concentration determined?**
- 10. What was the concentration used for iridium intercalator?**
- 11. For each antibody panel, please indicate intracellular and extracellular markers/targets (and catalog number) for reproducibility purposes.**
- 12. How was viability accounted for in some of the experiments? For example, in the monocyte isolation and glucose experiment, there is no indication regarding cell viability values, whether it varied between samples, and if so, what measures were taken to adjust it so it will be equal for all groups.**
- 13. For "Human brain immune cell isolation", can the authors comment on how relevant are samples taken 25 hours post-mortem? Is there any indication that this time period affects (or not) cell phenotype?**
- 14. Please elaborate more regarding cell fixation details in the methods section for reproducibility purposes.**
- 15. How were PBMCs isolated in this study?**

Reviewer #1 (Remarks to the Author):

This manuscript from Zapata et al. titled, "Differential compartmentalization of myeloid cell phenotypes and responses towards the CNS in Alzheimer's disease," explores immune cell phenotypic differences between the blood and cerebrospinal fluid (CSF) in healthy adult patients and those with Alzheimer's disease (AD). The authors provide data showing phenotypic differences in immune cell populations between the blood and CSF in both control and AD patients. The author's applied numerous state-of-the-art techniques to probe immune cells in these compartments and have generated an impressive dataset. However, although there may be potentially interesting results, the manuscript is overwhelming and does not highlight key findings. The figures are overcrowded and the results at times are overinterpreted since data are mostly descriptive and functional characteristics and consequences remain unexplored. Below are specific key points for each figure that should be addressed.

Response:

We very much appreciate the reviewer's critics and comments, which are very helpful for a significant improvement of our manuscript. We have re-structured the figures and main text, for being less overwhelming but still provide all necessary information. We also agree with the reviewer that this kind of studies (in human system) are mostly descriptive and an in-depth functional characterization is highly challenging, as we already have discussed in our original manuscript. However, we do trust that these datasets will provide an important resource for further studies. Nevertheless, we have also performed further experiments for a better characterization of functional changes and added into this revised manuscript. These should provide more information about changes in function of blood myeloid cells (at least responses to stimulation) once they are exposed to the CSF.

Figure 1:

• Several data plots are shown for each analysis, but most are glossed over without discussion or explanation. The authors should discuss each plot shown if it is to be included. For example, Fig. 1B shows NRS values for several genes, but these data are better expressed in Fig. 1D,E, which shows the expression of these variable genes in different cell clusters.

Response:

We apologize for the unclear description. We use the NRS values to determine the markers used for embedding and so we deemed important to include this information in the figures. As it was described in details in the original manuscript (page 5), we choose the top ten markers with highest NRS values (potentially the main markers that determine the differences between the cell clusters) for clustering analysis. This will give a robust and reproducible clustering analysis. Of note, the NRS (Fig. 1B) and expression levels after clustering analysis (Fig. 1D and E) do not provide the same information, as NRS instead is useful to get an idea of at which degree each marker is contributing to the variability in each sample.

After clustering, we can clearly see these phenotypic differences in Fig. 1D, E, confirming the reliability of the top ten NRS high markers. We rewrote the text to make this point more understandable.

• The terms “TYPE” and “STATE” markers are used throughout the manuscript but are never defined.

Response:

In the original manuscript, we wrote (page 5):

We selected the top ten highest NRS and lineage markers were selected as embedding (“TYPE”) markers, thus used as input for the clustering, while the rest of the markers were left as “STATE” markers.

In the revised version, to better describe the differences between the two we have revised the sentences to (see page 5 in the revised manuscript):

To achieve a robust phenotypic differentiation between the single cells, we selected lineage markers and the top ten highest NRS markers (Fig. 1B) as input (i.e. embedding markers) for the clustering analysis. These markers (here referred to as “TYPE” markers) mainly determined phenotypic differences between the cell clusters. The rest of the markers were left as “STATE” markers, which were then used to analyze differential marker expression of each cluster between conditions.

• Fig.1C is not referenced properly within the text.

In the original manuscript, we wrote:

This phenotypic variance could be explained by differential expression of for example CD3, CD14, MRP14, CD8a, CD4, CD61, CD11c, CD35, CD38 and CCR5 as shown by the MDS-based non-redundancy score (NRS)²³ (Fig. 1b,c).

In this revised manuscript (page 5), we rewrote to:

This phenotypic variance may mainly be explained by differential expression of CD3, CD14, MRP14, CD8a, CD4, CD61, CD11c, CD35, CD38 and CCR5 as shown by the MDS-based non-redundancy score (NRS)²⁵ of each sample (Fig. 1B). Differences in cell compositions between CSF and blood can be illustrated in the UMAP plot (Fig. 1C).

• Measures of “proportion” are misleading. For example, the authors claim, “The CSF-enriched clusters were mainly identified as myeloid cells (Clusters 13 and 15);” however, Cluster 13 was only expressed in two samples. Moreover, these clusters are not easily visible on the UMAP plots. Are these “proportions” being driven simply by a significantly lower number of total cells in the sample? Raw cell yields for each sample and resulting clusters should be reported.

Response:

Regarding the cluster 13, we thank the reviewer for this critic and agree that this may lead to misinterpretation. Although it was statistically significant, we mention now in the text that this result needs to be interpreted with caution. We also add now the information of cell numbers per sample and cluster and further proportion in the resulting clusters (**rev. Fig. 1H** and **Supplementary Fig. 1C**). Of note, now in the revised manuscript we have included all groups (i.e. CON, MCI, AD and HD) in Fig. 1. The proportion of the differentially abundant cluster 13 remains significantly higher in CSF (**Fig. 1F**). Also, we do think that the “proportions” are mainly driven by cell population detected in the samples. And since some populations such as B-cells are missing in the CSF, the “proportions” of myeloid cells in total cells will be higher. This was

also the reason why we show “proportions” rather than absolute cell numbers to avoid misinterpretation according to large differences of cell numbers between compartments.

• *“In addition, a differentially abundant cluster CD4+ T cell (Cluster 17) was detected at a higher frequency in the CSF.” The term “higher frequency” is misleading since the raw numbers are significantly lower in the CSF.*

Response:

To avoid any misinterpretation, we reworded from “higher frequency” to “higher proportion”.

• *Actual phenotyping results are presented in passing. “Overall, compared to classical monocytes (Cluster 16), CSF-enriched myeloid cells showed higher expression of markers involved in inflammatory responses, phagocytosis and metabolism... whereas MRP14, CD14, CD35, EMR1, CD38, CD369 (Clec7A) and TNF were expressed at a lower level.” These seem like the results that should be discussed more in-depth here. Additionally, TNF is reported to be expressed at a lower level, but “inflammatory responses” are reported to be higher. This warrants discussion.*

Response:

We apologize for this unclear description. Actually, we meant higher expression of markers involved in inflammatory responses, which did not mean higher expression of inflammatory markers or higher inflammatory responses. We agree with the reviewer that this may lead to misunderstanding. Therefore, we carefully reword some parts of the manuscript from inflammation to activation.

• *In panel 1G, it seems like there are more samples for PBMC-CON than CSF-CON. They should be equal (N=11) since taken from the same individual. Were samples excluded?*

Response:

We have now addressed the criteria of sample exclusion in the reporting summary and in the method section (page 32 & 33): only samples with more than 50 cells were considered for downstream data analysis.

Figure 2:

• *The authors aim “to prove the existence of this rare cell population,” referring to the “rare population of myeloid cells in the CSF with a transcriptomic signature matching microglia,” which are reported to be “found only in the CSF of subjects with neuroinflammation,” by performing “another CyTOF measurement of the same CON donors.” Why are controls being used if this is an inflammatory population? Also, is this population not CD16+/CD14 low non-classical monocytes which have been shown to be enriched in CSF compared to blood?*

Response:

This rare population was previously proposed by others as a microglia-matching cell population, that were found only in the CSF of subjects with neuroinflammation. However, as written in the original manuscript, our hypothesis is different. We hypothesize that this population is not a microglia-matching cell population and will exist independently to neuroinflammation, but their existence is rather influenced by changing environment (e.g. upon

entry the CSF from the blood). Entering the CSF, myeloid cells will change their phenotypes. Among these phenotypic changes is the expression of P2Y₁₂, a microglia marker that was widely used to distinguish brain microglia from other CNS macrophages. Of note, some peripheral macrophages have also been shown to express P2Y₁₂ (Chen et al. Mol Ther 2022). On the basis of our results, these cells belong to the circulating belong to myeloid cell population and we prefer not to call them microglia-matching cells. To avoid misinterpretation, we have rewritten this point and make it clear how we propose and hypothesize. In addition, the new in vitro experiment performed during the revision process also confirm an increased proportion of P2Y₁₂⁺ cell population or increased expression of this marker once blood myeloid cells were treated with CSF and/or both CSF and LPS (**revised Fig. 4 and 5**)

• Though the “CSF-enriched” cells discussed can be identified on the UMAP plots, they do not seem to cluster separately based on expression patterns.

Response:

We agree with the reviewer that their phenotype overlap with blood myeloid cells. We think that there must be more markers (still unknown for us) required for being able to separate this population from other myeloid cells in the peripheral blood. A following and on-going study in which we have additionally measured over 120 CSF samples (from different diseases) have confirmed this finding but we are not yet successful to identify missing markers required for further characterization of this population.

• Additional microglia signature markers should be assessed (e.g., TMEM119, SALL1, TGFBR1) before determining these cells are “matching microglia”.

Response:

As mentioned above, we did not claim that this population is “matching microglia”. We consistently name this population “myeloid” in purpose. To avoid misunderstanding, we have re-written the text. We use here P2Y₁₂, which has been widely used as a microglia signature marker, in combination with CCR2 and CD16 (**revised Fig. 2, 5 & 6**).

Figure 3:

• Are the clusters in Fig.3 the same as in the previous figures? If so, why are they being compared separately from CON? These should have CON in the same figure as reference to evaluate changes with disease condition. If not, how are these clusters being generated, and how can they be compared to the previous clusters?

Response:

We thank the reviewer for pointing this unclear issue out. Yes, they are the same as in the previous figures. We have now re-arranged all the figures again and combined the CON and the other disease condition groups (**revised Fig. 1 and 2**).

• The authors conclude there are “different abundances in myeloid, NK and lymphoid cell clusters in CSF compared to the peripheral blood”; however, the setup of Fig.3B divides the analysis into disease state (i.e., MCI v. AD v. HD). These distinctions are not discussed in the text.

Response:

We thank the reviewer for pointing this missing information. As mentioned above we have now rearranged the figures to show the comparison between disease conditions and the CON group, and we have revised the main text accordingly.

Figure 4:

- *IL-10 is introduced in text, but all other references claim it was IP-10.*

Response:

We did measure both IL-10 and IP-10. Since we did not observe any differential regulation in IL-10, and the level of plasma and CSF IL-10 were under limit of detection in all samples analysed, we did not mention this later in the text. To avoid any misunderstanding, we have now revised the text accordingly (page 9).

- *Confusing text: “The Luminex assay revealed a higher concentration of IL-8, MIP- β , CCL2 (MCP-1), IL-6 and IP-10 in the CSF, whereas the level of the plasma TNF and Rantes (CCL5) were higher than in the CSF.”*

Response:

We apologize for the confusing description. In this revised manuscript, we have re-worded the sentence (page 9).

- *Labels for Fig.4C need work. Y-axis should read, “CSF MIP- α Concentration (pg/mL)” or “CSF IL-6 Concentration (pg/mL)”; X-axis should read, “CSF IL-8 Concentration (pg/mL)”; and Legend should read, “CON”, “MCI”, “AD”, and “HD”.*

Response:

We thank the reviewer for pointing this out and apologize for the mistake. We have re-worded as suggested (**revised Fig. 4**).

- *Why are data for FTD and SCZ not presented?*

Response:

We apologize for the missing information. Now, we have included all groups in the figure (**revised Fig. 4**).

Figure 5:

- *The biological relevance of ex vivo LPS treatment is not clear. First, how is CSF being isolated/applied? Is the CSF depleted of cells? Moreover, LPS will seldom-if-ever directly stimulate cells in CSF. Another more biologically relevant assay should be employed here.*

Response:

We have used cell-free CSF, thus the effects driven by LPS stimulation of CSF cells can be excluded. This missing information is now included in the method section.

We agree with the reviewer that it is very challenging to set-up a biological relevant ex vivo analysis, when the causal trigger(s) remain unknown. The reason why we use LPS was that

1. It is unknown which factor(s) in the CSF would drive the phenotypic, metabolic and/or functional changes of myeloid cells. Since we did not detect changes in canonical activators of myeloid cells such as CCL2, IFN α , IL-6, IL-10 that act through specific

receptors. And it is known that in the CSF $A\beta_{42}$ in AD is lower than in the control individuals, thus $A\beta_{42}$ may not be a good stimulant for circulating immune cells. Moreover, since we focus on myeloid cell, we decided to use a powerful stimulant that activates myeloid cells via Toll like receptor 4 (TLR4) such as LPS, to determine myeloid cell responses to inflammatory cues, as a difference in responses (e.g. phenotypic changes and cytokine production) may relate to the pathology.

2. Secondly, we would like to assess the synergistic effects of CSF and second stimulation such as LPS. Whether or not, cells that exposed to CSF are more vulnerable to secondary trigger like LPS.

During the revision, we have also performed stimulation experiments, using IL-8 and MIP-1 α , on the basis of our finding that CSF levels of these two molecules were higher in AD-CSF compared to other conditions shown in **Fig. 4** (in the original and revised manuscript). However, we could not detect gross effects of both cytokines on phenotypic changes of myeloid cells in AD. Nevertheless, a functional validation in an animal model is helpful but may not fully mirror the pathology in humans especially when investigating slowly expanding neurodegeneration and/or neuroinflammation.

• Which CSF is used in “CSF-only” and “CSF+LPS” treatment? The text claims both CON-CSF and AD-CSF are used. Are these mixed?

Response:

We have revised this part for better understanding, and separate the AD-CSF from AD-CON (**revised Fig. 6, page 11 and 12**). The experiment was performed with CON-CSF and AD-CSF separately but we analyzed the data together to assess a possible general CSF effect.

• The confusion introduced by the two points above make interpretation of the rest of the figure difficult if not impossible.

Response:

We thank the reviewer for this critic. We have now revised the manuscript to avoid the confusion (**page 11 and 12**).

• More work would be needed (not just relying on CCR2 and P2RY12) to conclude Cluster 15 is similar to the population in fig 2.

Response:

We thank the reviewer for this comment. Among all 36 markers used in **Fig. 2** (which makes at least 630 different combinations of 2 markers), we observed a meaningful correlation of CCR2 and P2Y₁₂ expression, which can be used to identify differences between blood and CSF-enriched myeloid cells (i.e. cluster 16 in **Fig. 2**). Therefore, we think that this combination is useful to identify similar cell subsets that can be induced after CSF exposure, and thus can be used to prove the influence of CSF on myeloid cell phenotype. However, based on the limitation of our current technical possibility we could not isolate the P2Y₁₂⁺ population shown in **Fig. 2** and directly compare them with the cluster 15 in **Fig. 5**. Furthermore, it has been shown and widely accepted that transferring endogenous myeloid cells (e.g. microglia) to the in vitro environment results in significantly changes in the expression of thousands of genes and possibly downstream protein expression (Gosselin, D. et al. Science. 2017). Therefore, the results need to be interpreted with caution, since we could not rule out the effects of the in vitro environment on cell phenotypes. In this revised manuscript, we have therefore reduced our interpretation and have added more discussion about this result in the main text.

Figure 6:

- Authors should use consistent terms between text and figures (e.g., pyruvate v. pyruvic acid; lactate v. lactic acid).

Response:

We have re-worded accordingly.

- For Fig.6E, does the decreased “ratio-pyruvate-to-lactate” not indicate increased lactate production (i.e., increased value of the denominator)? The text seems to interpret these data backwards.

Response:

We apologize for this confusion, and have re-worded “ratio” to “conversion” of pyruvate to lactate, which was calculated by concentration of resulted lactate/divided by concentration of pyruvate.

Figure 7/8:

- Fig.8E is thrown into the middle of the discussion of Figure 7. Moreover, this is the only panel discussed for Figure 8. Authors should discuss all data presented in figures.

Response:

We thank the reviewer for this helpful critic. To avoid unnecessary confusion, we have removed the Fig. 8. And to simplify the illustration of the results, we divide the Fig. 7 into Fig. 7 and 8.

- Discussion of these data ends on a very weak note: “Together, it is tempting to speculate...”

Response:

We reworded the discussion, and put more strength on our finding but avoided overstated conclusions.

General:

- The authors repeatedly write, “We asked whether... To prove this assumption...” but no assumption is being clearly made. Changing this to, “To explore this...” or another phrase may make the thought process of the experimenters easier to follow.

Response:

We reworded the manuscript accordingly.

- The figures are overcrowded

Response:

We have simplified our figures but also shown/illustrated sufficient data allowing readers to judge our study and to keep consistency and transparency.

Reviewer #2 (Remarks to the Author):

The authors Zapata et al., have presented an interesting study on the myeloid cell phenotypes in healthy and AD individuals. However, the study needs some additional analysis and clarification that are suggested here:

1. The authors have provided donor and patient information. But details like the number of males and females in this study, APOE status of the individuals, Braak or CERAD, or cognitive

assessment scores are not mentioned. The authors should include this information in the text as well as the supplementary.

Response:

We thank the reviewer for this comment. In the original manuscript, we have already addressed the number of males and females in this study. As suggested, we also have now added more clinical information, which are available, into the supplementary information. We have addressed the Braak scores only for post-mortem samples. We do not have this information in living patients. In addition, correlation analysis of the finding with age is now included (**revised Figure 4**).

2. Age as well as sex influence the neuroinflammatory response. The authors should include analysis by stratifying the samples based on the age and sex of the individuals considered in this study.

Response:

We have added the correlation between age and cytokine level in the CSF in the revised manuscript (**revised Figure 4**). Gender differences in IL-8 and MIP-1alpha levels in the CSF was now shown in **revised Supplementary Fig. 3**.

3. Can the authors describe more about the TYPE and STATE markers? How are they different? The median scaled expression of STATE markers in both Figures 1 and 2 is lesser than TYPE markers. So, is there a significance of STATE markers if we consider the expression values?

Response:

In the original manuscript, we wrote:

We selected the top ten highest NRS and lineage markers were selected as embedding ("TYPE") markers, thus used as input for the clustering, while the rest of the markers were left as "STATE" markers.

In the revised version, to better describe the differences between the two we have revised the sentences to (see page 5 in the revised manuscript):

To achieve a robust phenotypic differentiation between the single cells, we selected lineage markers and the top ten highest NRS markers (Fig. 1B) as input (i.e. embedding markers) for the clustering analysis. These markers (here referred to as "TYPE" markers) mainly determined phenotypic differences between the cell clusters. The rest of the markers were left as "STATE" markers, which were then used to analyze differential marker expression of each cluster between conditions.

4. How much was the difference in the cell numbers in CSF and PBMC? Based on Figure 1C, PMBC has more clusters and cells than CSF. Was cell count normalization carried out before comparing the markers between the groups?

Response:

During the data processing and analysis, we actually performed down-sampling (cell count normalization). However, these data were not shown in the original manuscript, due to the space limitation and for the sake of readability. Furthermore, since the results we have obtained from down-sampling strategy were not different from the analysis using total cells (without cell count normalization), we decided to show the results from the analysis with all cells that we have detected in the samples. We believe that this information will give a better overview for the readers of cellular composition in the blood and CSF in total.

The following results were obtained from down-sampling analysis (not shown in the manuscript):

Analysis of Panel 1 PBMCs and CSF cells downsampled to max. 850cells/sample

The Figure shows the results of data analysis using Panel 1 and the same workflow as was used in the manuscript, comparing the CSF and blood cells. **A)** MDS plot, **B)** NRS plot, **C)** UMAP plot of overlaid data from all samples, **D and E)** UMAP plots of defined clusters, **F)** Heatmap showing the phenotypes of 20 defined clusters and **G)** Bar graphs showing differentially abundant clusters between the two compartments.

5. Lipid species have been studied in detail as they are associated with neuroinflammatory responses. Did the authors identify lipid markers in their study?

Response:

In this manuscript, we did not specifically assess lipid markers. However, we agree with the author that lipid species have been reported to be involved in neuroinflammation. We have two ongoing projects in which we specifically characterize lipid markers in neuroinflammation. However, these projects have not yet been completed.

6. In figures 3 and 4, the differential marker expression is similar in AD and MCI samples. Does it suggest that the neuroinflammatory response is similar in both disease phenotypes?

Response:

We thank the reviewer for this interesting comment. To assess more similarity and differences in inflammatory responses between these two conditions, we have performed additional *in vitro* experiments including MCI PBMCs (**revised Figure 4 and 5**).

7. *Can the results presented here use for building predictive models for the onset of AD?*

Response:

As we have mentioned in the discussion, it is still too early to use the findings in this study for prediction or monitoring AD phenotypes. But our study strongly suggests that to study or understand neuroinflammatory responses (for example in AD) it is necessary to assess immune responses in different body compartment with different technologies/methodologies to evaluate various changes including metabolism (e.g. using Seahorse), phenotype (e.g. using CyTOF) and function/response (e.g. *in vitro* stimulation assay). For example, knowing more about changes in immune cell metabolism in different body compartments would give also new insights or new strategies for treatment.

8. *In Supplementary Table 1, samples for depression, FTLN, and SCZ are indicated. Have these been used for comparative analysis?*

Response:

We apologize for this confusion. We have now added more description about these conditions in the main text and also in the Figure (revised Figure 4).

9. *"Venous blood and lumbar cerebrospinal fluid (CSF) samples were obtained from control individuals or patients with neurological disorders (Supplementary Table 1)." Were the samples collected only once or more than one time from these individuals?*

Response:

The samples were collected only one time.

Minor comment:

In the abstract, kindly rephrase "health" to "healthy"

Response:

We have reworded, as suggested.

Reviewer #3 (Remarks to the Author):

Fernández Zapata and colleagues provide a detailed analysis of multiple human compartments and cell types with various high-dimensional technologies. In their study, they comprehensively characterized human-derived samples of blood, cerebrospinal fluid, choroid plexus, and brain parenchyma while focusing on immune cell abundance and phenotype. The authors used several algorithms to analyze the generated data and interpret it. It is important to appreciate the extent of work done in this study due to the use of human samples and their limited availability, especially those derived from the central nervous system. As the authors mentioned, using human systems is ethically and technically challenging.

While the manuscript is intriguing, the majority of the data is descriptive, and some technical and conceptual factors are missing, along with some issues in data interpretation. Additionally, data describing the composition of human blood, cerebrospinal fluid (CSF), choroid plexus,

and brain parenchyma are available in current literature (with same or other analysis methods; PMID: 33239300, PMID: 33239300), thus limiting the novelty of this study. Moreover, the main conclusions are mostly hypothetical, as also stated by the authors. Collectively, the authors do provide a comprehensive depiction of cells and secreted factors, at the protein level, between different compartments, yet they mostly rely on one method (mass cytometry) and no validations were used for key results.

Response:

We thank the reviewer for the comment. We agree with the reviewer that studies on immune composition in human blood, CSF, choroid plexus or brain are already available. However, to our knowledge, there is thus far no study that comparatively characterizes immune cells in four different compartments using high-dimensional single-cell mass cytometry, *in vitro* stimulation assay and metabolic analyses (i.e. Seahorse and ¹³C-glucose-tracing experiment). Therefore, we do believe that our study is novel and state-of-the-art. In the paper suggested by the reviewer *PMID: 33239300*, in which only CyTOF was used to comprehensively assess phenotypic and functional profiles of only AD PBMCs, this shows also the limitation of possibility of performing analysis using human specimens, as also mentioned by the reviewer. Nevertheless, in our study, in addition to those known cytokine/cell signalling pathways regulated during *in vitro* stimulation, we also demonstrated changes in metabolism especially glucose metabolites, which is important in myeloid cell function. This is novel and could be an important additional information to, for example, findings shown in *PMID: 33239300*.

Major comments:

1. Comparing different cell clusters, meaning different cell subsets, will most likely provide significant differences between the clusters. Results describing cluster-specific phenotypes, although interesting and valuable to understand the nature of each cell subset, should not take so much focus. It is the group-specific differences that need to be thoroughly addressed. It is also somewhat confusing to show significant differences between clusters along with other group-based analyses.

Response:

We agree with the reviewer and have added more analyses of group-specific differences (**revised Fig. 1 and 2**). However, as also recognized by the reviewer, we would like to keep the cluster-specific phenotype analyses, since this information gives us better insights of phenotypic signatures of differential abundant clusters in each condition.

2. Tissue-related comparisons are another similar example of results that are important to understand the cell environment in each compartment, yet should be used for supporting the main data rather than being it. For example, besides further validating that CSF and plasma have different compositions, what does the data in Figure 4a provide us? It would be more interesting to see the levels of each inflammatory mediator for each group (control, AD, MCI, HD, FTL, depression and schizophrenia), including those that were not included in Figure 4b.

Response:

We thank the reviewer for this suggestion and comment. We have now included the results of all groups in the **revised Figure 4**.

3. An important variable in this study that is not referred to by the authors is donor age, some of the differences may be age-related, as the mean age for the CON group is 62 yr while in the

AD group it is 72 yr. That 10 year gap may be a significant contributing factor of inflammation. Furthermore, if possible, please also refer to BMI and other relevant parameters of donors. On the same note, are there any gender-specific differences that could be identified in the various compartments?

Response:

We have now added age-related changes in cytokine profiles (**revised Figure 4**), as well as gender differences in IL-8 and MIP-1alpha levels in the CSF (**revised Supplementary Fig. 3**). BMI data is not available in this study.

4. It was confusing and laborious to go back and forth with several of the figures due to the split layout according to panel. Might be better to have the data side by side according to plot type and indicate the relevant panel below each part or in the figure legend.

Response:

We thank the reviewer for this valuable comment and suggestion. We have now rearranged the figures by dividing the original Fig. 7 into Fig. 7 (for Panel A) and Fig. 8 (Panel B). We found that directly compare the two panels in one figure may not improve the readability but rather make it more confusing.

5. While the manuscript is mostly well-written (with minor typos), the results section contains too many technical notes (e.g., page 5). It would benefit the reader to pare out as much of this as possible. This information can be included in figure legends and methods, and thus the paper will be much more readable. Negative data could be deemphasized by significantly shortening these sections.

Response:

We thank the reviewer for this suggestion. The manuscript has been revised as suggested. However, based on the comments of the reviewer #1, some technical information are required for better readability.

Minor comments:

1. The manuscript is lacking gating examples for CyTOF gating (e.g. live single-cell gating). Also, manual gating to some of the results would provide some reassurance to the results generated by some of the algorithms/packages; some results rely on a very limited number of cells and their inference might then be revised (for example, cluster 18 in Figure 4d). Accordingly, referring to cell counts (per subset/analysis) would support the validity of the data. Some CSF subsets/clusters likely have less than 10 cells; is it possible that some of the CSF-PBMC cluster proportion differences were created by the large difference in total cell counts between these compartments? (perhaps downsample). Please also provide a gating example for cell sorting according to tissue.

Response:

We have now added the gating strategies in the revised **Supplementary Figure 1, 4 and 5**. We have proven the abundance of rare populations by manual gating, as suggested by the reviewer. Please note there was no "cluster 18 in Figure 4d". This may be just a typo, since there was no clustering analysis in Fig. 4. Therefore, we show here an example of manual gating for CSF-enriched cluster 16 (**revised Fig. 2**), which is one of our main findings:

As shown in the figure, we could confirm the abundance of $HLADR^+CD11c^+CD68^+CD91^+CCR2^{low}/P2Y_{12}^+$ and the similar cluster (cluster 15), which is $HLADR^+CD11c^+CD68^+CD91^+CCR2^+P2Y_{12}^{low}/-$. However, we have decided not to include these results for all rare populations defined in this study, in order to keep readability of the manuscript, unless it is required.

As response to the Reviewer #2 (see page 9 of this document), during the data processing and analysis, we actually performed down-sampling (cell count normalization). However, these data were not shown in the original manuscript, due to the space limitation and for the sake of readability. Furthermore, since the results we have obtained from down-sampling strategy were not different from the analysis using total cells (without cell count normalization), we decided to show all cells that we have detected in the samples. We believe that this information will give a better overview for the readers of cellular composition in the blood and CSF in total.

2. Many plots indicate “Expression” on the Y-axis with no units, please clarify. Can the authors elaborate on what is a reasonable expression level? For example, does the small difference in expression level (~ 0.1) in Figure 7f Cluster 4 CP has a biological meaning? Please explain for other plots with low expression values.

Response:

As the signal intensity obtained from CyTOF measurements is generally much lower than the signal intensity from flow cytometric analysis (by factor 10 - 100), and to our experience a hyperbolic arcsine (arcsinh)-transformed value about 0.1 can be used as a reliable value for marker expression. However, we can't conclude or claim that this difference has a biological meaning. To prove a biological meaning of a protein, a model that particularly and selectively manipulates this protein (such as mouse models of knock-out or over-expression of the protein) in diseases is required. This is however beyond the scope of this paper.

3. Please indicate “Proportion (%)” out of what population/pool of cells (i.e., all CD45+ live single cells?).

Response:

Proportion (%) is of total CD45⁺ cells. We have added this missing information in the figures.

4. Some of the presented UMAPs lack the total number of cells and the number per condition/tissue. This is especially important in CSF samples that in some cases include very few cells.

Response:

We apologize for missing this information in some of the UMAP plots. We have now added the requested information in the figures.

5. It would benefit the reader to have cluster annotations in each heat map, along with cluster numbers over each dimensionality reduction map (color blind compatible).

Response:

We have revised the figures as suggested.

6. Please clarify the exact details of “cell culture and stimulation”. For example, how was the volume of CSF determined in the in vitro experiments?

Response:

Since CSF has very low buffer capacity and in a large volume can dramatically change the pH of the medium, we used the maximum volume of CSF that give a maximum effect but does not change the pH of the cell culture. We use 20% CSF. This volume has been tested and used for Seahorse, ¹³C-glucose tracing and *in vitro* stimulation assays.

7. For Figure 5, the authors mention that “PBMCs showed changes in phenotype when treated with CSF”, however, the MDS plot does not show much difference between no stimulation and CSF. Small if any differences are also evident in Figure 5c. This lack of difference is also surprising compared to the results in Figure 6, showing a significant change when CSF is added.

Response:

We thank the reviewer for pointing out this unclear description. We have revised the main text to avoid any over- or mis-interpretation. As mentioned by the reviewer, according to the MDS plot, we did not observe any overall differences between no stimulation and CSF-treated conditions. Only small differences in the abundance of some myeloid cell subsets and rare mixed population can be detected (Fig. 5c in the original manuscript). However, the MDS plot showed overall gross differences once the CSF-treated myeloid cells were also stimulated with LPS (i.e. LPS-treatment versus CSF+LPS treatment). We have also repeated this experiment using PBMCs isolated from CON, MCI and AD individuals (**revised Figure 5**), instead of using healthy PBMCs as in the original manuscript (**revised Figure 6**). The phenotypic differences between groups observed from this experiment were more pronounced (**revised Figure 5**). Furthermore, one of the main differences between Seahorse and CyTOF-in vitro experiment is the incubation and time point of analysis. The metabolic changes were already detected shortly after CSF exposure using Seahorse (i.e. about some minute after CSF injection and lasted more than 2h after injection), whereas CyTOF measurement was performed after 6 hours incubation.

8. In Figure 8, since all pooled samples were not analyzed in the same CyTOF run, how was signal intensity normalized between samples/runs?

Response:

For this experiment, the measurements were designed using standard calibration beads for normalization of signal intensity within the runs. However, due to the comment of reviewer #1, we have now removed this figure from the revised manuscript, since it doesn't give any additional information and is rather confusing.

9. In the in-house generated CyTOF antibodies, how was optimal antibody concentration determined?

Response:

Each antibody was titrated and validated as into the working panels prior to use to ensure that the resulted signals were informative. The titration was performed using different cell types (i.e. PBMCs, stimulated PBMCs and isolated CNS myeloid cells), as in our previously published papers (Böttcher et al., Nat. Neurosci. 2019; Böttcher et al., Sci. Rep. 2020).

10. What was the concentration used for iridium intercalator?

Response:

We used the concentration of 500 nM intercalator.

11. For each antibody panel, please indicate intracellular and extracellular markers/targets (and catalog number) for reproducibility purposes.

Response:

Now we have added the missing information.

12. How was viability accounted for in some of the experiments? For example, in the monocyte isolation and glucose experiment, there is no indication regarding cell viability values, whether it varied between samples, and if so, what measures were taken to adjust it so it will be equal for all groups.

Response:

Viability was either accounted using classical trypan blue staining. For consistency of the experiment, we control the number of cells seeded into each well to be equally between conditions. The viability (trypan blue staining) was non-significant between conditions studied.

13. For "Human brain immune cell isolation", can the authors comment on how relevant are samples taken 25 hours post-mortem? Is there any indication that this time period affects (or not) cell phenotype?

Response:

In our study, we control the post-mortem delay to be less than 10 hours. This refers to the time from death to brain isolation. The brain tissue is collected in the operating room, transported and processed in cold carbogenated NMDG-aCSF (95% O₂, 5% CO₂) containing (in mM): NMDG (93), KCl (2.5), NaH₂PO₄ (1.2), NaHCO₃ (30), MgSO₄ (10), CaCl₂ (0.5), HEPES (20), glucose (25), sodium l-ascorbate (5), thiourea (2), sodium pyruvate (3), as described previously (Böttcher et al. Nat. Neurosci. 2019). The protocol was validated to provide reliable results on cell phenotype, when the cell isolation is started within 2 to 25 h after autopsy (Mizee et al., Acta Neuropathol Commun. 2017; Mizee et al, Handb Clin Neurol. 2018). In addition, it has been shown that the pH of the CSF is correlated to the yield and the quality of isolated

microglia (Mizee et al., Acta Neuropathol Commun. 2017) Therefore, in this study the brain tissue was collected only from the donors whose postmortem CSF was between pH 6 and 7.

14. Please elaborate more regarding cell fixation details in the methods section for reproducibility purposes.

Response:

We have now added this information in the method section (page 30).

15. How were PBMCs isolated in this study?

Response:

We apologize for the missing information. We have now added this information in the method section (page 30).

REVIEWER COMMENTS

Reviewer #1 (Remarks to the Author):

Overall, the authors greatly improved the manuscript however there are still concerns.

- The figures are still very dense and superfluous analyses between cells in different tissues (i.e. blood and CSF or CP and brain), distract from the more important analyses between disease states.**
- I'm still not convinced that "neuroinflammation-associated microglia-like cells" are not just a population of non-classical monocytes.**
- The in vitro studies are confusing. Since the author's initial conclusions are that PBMCs isolated from AD patients are more sensitive to in vitro experimental conditions, how can anything be concluded from then adding LPS? In general, the rationale for using LPS as second hit is weak in regards to relevance.**

Minor issues:

- Figure 1 adjust the scale. Looks like the number of cells in each dot from CSF is well below the smallest 40,000 scale dot.**
- Figure 1D, 2B, 6A, etc.: Why are only some clusters numerically labeled?**
- Fig 6A. Doesn't look like 20 clusters**
- "Myeloids" should not be plural.**

Reviewer #2 (Remarks to the Author):

The authors have revised the manuscript and updated the figures to include additional information. The revised version of the manuscript addresses all the comments. The work presented by the authors is important in the AD field.

Reviewer #3 (Remarks to the Author):

The authors have provided substantial changes to the manuscript and have adequately addressed most of my comments.

As mentioned before, data are mostly descriptive, and, as now shown, cell counts are relatively limited in some analyses.

Nonetheless, these data would be valuable to the scientific community and could support other studies.

While referring to prior comments (reviewer #3):

Minor comment #2, for the justification of a biological meaning of such small differences in intensity, the authors could refer to other works that showed similar changes along with functional outcomes, and refer to those in the manuscript. Note that the reliability of the arcsinh-transformed value of about 0.1 depends on many factors, including the used conjugated metal and antibody.

Minor comment #4, please add cell counts in the UMAP of Figure 6A.

Minor comment #8, please refer to normalization/harmonization of data from separate/different runs. If I understand correctly, all data were derived from the same mass cytometry or flow cytometry run, and all cells were cryopreserved beforehand. Please clearly indicate this in the manuscript.

Minor comment #10, please indicate the concentration within the manuscript for reproducibility.

Reviewer #1 (Remarks to the Author):

Overall, the authors greatly improved the manuscript however there are still concerns.

- The figures are still very dense and superfluous analyses between cells in different tissues (i.e. blood and CSF or CP and brain), distract from the more important analyses between disease states.

Response:

We thank the reviewer for the positive feedback regarding our efforts to improve the quality of the manuscript.

As we have mentioned, one of the study aims is to characterize compartmentalization of myeloid cells circulating or residing in different body compartments. We hypothesize that changes in myeloid cell compartmentalization may associate with pathology observed in neurological diseases at different compartments, including CSF and brain barrier. Therefore, demonstrating phenotypic changes across body compartments in details is required for our story. We also convince that it will also be beneficial for the readers outside the Alzheimer research field, who are interested in changes of immune cell compartmentalization across body compartments that may associate to neuropathology. However, we did follow the suggestion of the reviewer and have simplified all figures and have shown only necessary results.

Together, we trust that the comprehensive immune profiles demonstrated in our study will be profitable for a broader spectrum of readers including also those interested in systemic changes in neurological diseases other than AD.

- I'm still not convinced that "neuroinflammation-associated microglia-like cells" are not just a population of non-classical monocytes.

Response:

As mentioned in the manuscript and the previous point-by-point letter, we have questioned the use of the term "neuroinflammation-associated microglia-like cells". Our findings suggest that these cells should be cautiously termed "neuroinflammation associated microglia-like cells", as these cells were also present in the CSF of healthy donors.

Compared to non-classical CD16⁺ monocytes, the CSF-enriched myeloid cells differ in the expression of several markers. For more clarity, in **the revised Figure 2F** we have now included a comparison between C15 (classical CD16^{lo/} monocytes), C13 (non-classical CD16⁺ monocytes) and C16 (CSF-enriched CD16⁺ myeloid cells). These three subsets were differently clustered (**Figure 2B**). We also described this comparison in the revised main text now (**page 7**).

However, please also note that the nature of our experiments does not allow us to draw further conclusions about the origin of this population, whether or not this population might originate from the non-classical monocytes is highly speculative. Moreover, due to limitation of the analyzed markers we could not completely separate a small population of C16 from C15 and C13, as already mentioned in the last point-by-point letter.

- The in vitro studies are confusing. Since the author's initial conclusions are that PBMCs isolated from AD patients are more sensitive to in vitro experimental conditions, how can

anything be concluded from then adding LPS? In general, the rationale for using LPS as second hit is weak in regards to relevance.

Response:

As explained in the first revision, we have also performed in vitro experiments using IL-8 and MIP-1 α which seemed to be more biological relevance since they were found increased in CSF of AD patients. However, we could not detect any significant differences in phenotypes or cytokine production, compared to other conditions. Now, based on our new results (as shown in the first revision) we convince that the effects of changing environment may be bigger than the effects of specific molecule. Interestingly, LPS stimulation revealed even more significant increase in some of phenotypic changes, such as P2Y₁₂ expression and TNF production shown in **Figure 5F** (page 11). We agree with the reviewer that LPS as a second hit may not be a real biological relevance that could be detected *in vivo*. However, it is interesting to show that AD-PBMCs especially the CD14⁺CD16⁺ myeloid cells are vulnerable to insults (e.g. LPS) or environment changes and can then present similar phenotype as the CSF-enriched myeloid cells (i.e. P2Y₁₂⁺ myeloid cells).

Minor issues:

- Figure 1 adjust the scale. Looks like the number of cells in each dot from CSF is well below the smallest 40,000 scale dot.

Response:

We have adjusted the scale as suggested (**revised Figure 1A**).

- Figure 1D, 2B, 6A, etc.: Why are only some clusters numerically labeled?

Response:

We have now labelled all clusters in all UMAP plots.

- Fig 6A. Doesn't look like 20 clusters

Response:

In **Figure 6A** many of the clusters account for a small fraction of all cells, therefore they are difficult to discern in the UMAP plot. In this experiment we have taken advantage of the “overclustering” approach in which we can obtain more resolution and is a good tool for discovery of small populations or “cell states” otherwise difficult to identify.

- “Myeloids” should not be plural.

Response:

We have reworded in all figures according to the reviewer suggestions.

Reviewer #2 (Remarks to the Author):

The authors have revised the manuscript and updated the figures to include additional information. The revised version of the manuscript addresses all the comments. The work presented by the authors is important in the AD field.

Response:

We thank the reviewer for this positive feedback.

Reviewer #3 (Remarks to the Author):

The authors have provided substantial changes to the manuscript and have adequately addressed most of my comments.

As mentioned before, data are mostly descriptive, and, as now shown, cell counts are relatively limited in some analyses.

Nonetheless, these data would be valuable to the scientific community and could support other studies.

Response:

We thank the reviewer for this positive feedback and his/her understanding for the nature of our study.

While referring to prior comments (reviewer #3):

Minor comment #2, for the justification of a biological meaning of such small differences in intensity, the authors could refer to other works that showed similar changes along with functional outcomes, and refer to those in the manuscript. Note that the reliability of the arcsinh-transformed value of about 0.1 depends on many factors, including the used conjugated metal and antibody.

Response:

We agree with the reviewer that drawing biological meaning of small changes in expression of some markers is challenging and we are therefore always careful in overstating such conclusions. Nonetheless, low intensity levels in some markers is a known challenge when using CyTOF technology (as for example compared to flow cytometry), as has been addressed, e.g. Iyer *et al. Front. Immunol. 2022* and Gonder *et al. Front. Immunol. 2020*. Therefore, we validated all antibodies using different cell types (i.e. using immune cells in the blood and the brain) to be able to distinguish low expression and background. Moreover, if applicable we always put markers with low intensity in channels with high sensitivity (i.e. roughly between 157-170), Takahashi *et al. Cytometry Part A 2016*.

Minor comment #4, please add cell counts in the UMAP of Figure 6A.

Response:

We have adjusted the figure as suggested by the reviewer.

Minor comment #8, please refer to normalization/harmonization of data from separate/different runs. If I understand correctly, all data were derived from the same mass cytometry or flow cytometry run, and all cells were cryopreserved beforehand. Please clearly indicate this in the manuscript.

Response:

In the paper we only compared data from same runs, we clearly separate data from different runs note Fig 1 and Fig 2 for instance are different runs and are shown separately, therefore no normalization was needed.

Minor comment #10, please indicate the concentration within the manuscript for reproducibility.

Response:

We apologize for this missing information. We have now added this information in the revised manuscript (**page 33**).

REVIEWERS' COMMENTS

Reviewer #1 (Remarks to the Author):

The authors have adequately responded to all the concerns.

Reviewer #3 (Remarks to the Author):

The authors have adequately addressed all comments.